Methods

# High-resolution analysis of bound Ca²⁺ in neurons and synapses

Elisa A Bonnin[1,2,3], Arash Golmohammadi[4], Ronja Rehm[1], Christian Tetzlaff[4], Silvio O Rizzoli[1,2,3]

**Calcium (Ca²⁺) is a well-known second messenger in all cells, and is especially relevant for neuronal activity. Neuronal Ca²⁺ is found in different forms, with a minority being freely soluble in the cell and more than 99% being bound to proteins. Free Ca²⁺ has received much attention over the last few decades, but protein-bound Ca²⁺ has been difficult to analyze. Here, we introduce correlative fluorescence and nanoscale secondary ion mass spectrometry imaging as a tool to describe bound Ca²⁺. As expected, bound Ca²⁺ is ubiquitous. It does not correlate to free Ca²⁺ dynamics at the whole-neuron level, but does correlate significantly to the intensity of markers for GABAergic pre-synapse and glutamatergic post-synapses. In contrast, a negative correlation to pre-synaptic activity was observed, with lower levels of bound Ca²⁺ observed in the more active synapses. We conclude that bound Ca²⁺ may regulate neuronal activity and should receive more attention in the future.**

## Introduction

Calcium ions (Ca²⁺) act as important second messengers in all cells, regulating vital aspects of cellular function from cell proliferation to cell death (Schwarz & Blower, 2016; Giorgi et al, 2018). In the nervous system, Ca²⁺ signaling plays a special role in the regulation of neuronal and synaptic activities. The influx of Ca²⁺ into the synaptic active zone triggers the exocytosis of synaptic vesicles, facilitating neurotransmitter release (Brini et al, 2014). This is a fundamental mechanism for synaptic transmission and is mediated in part by the Ca²⁺ sensor synaptotagmin (Bacaj et al, 2013).

However, the Ca²⁺ ions that perform these functions are only a small fraction of the total amount of Ca²⁺ in a cell. Intracellular Ca²⁺ is divided into two pools: the free Ca²⁺ pool and the bound Ca²⁺ pool. The free Ca²⁺ pool consists of Ca²⁺ ions available to participate in chemical reactions, whereas the bound Ca²⁺ pool consists of Ca²⁺ ions bound to cytosolic buffer proteins, such as parvalbumin, calbindin, and calretinin (Fairless et al, 2019), or sequestered in

organelles as the endoplasmic reticulum (Schwarz & Blower, 2016) or the mitochondrial matrix (Vaccaro et al, 2017; Devine & Kittler, 2018; Giorgi et al, 2018). The vast majority (>99.9%) of intracellular Ca²⁺ is in the bound form (Neher, 1995).

Despite the difference in size between the free and bound pool, free Ca²⁺ has received much more attention over the last few decades, as this is the functionally active population. A variety of techniques have been employed to image free Ca²⁺ in the cell, mainly involving the use of fluorescent dyes or genetically encoded calcium indicators (Tian et al, 2012; Lin & Schnitzer, 2016; Ali & Kwan, 2020). In contrast, bound Ca²⁺ has received less attention. This is partly because of the lack of imaging methods for the bound Ca²⁺ pool. Unlike free Ca²⁺, bound Ca²⁺ cannot be imaged using fluorescence because the functional principle of all Ca²⁺ dyes or indicators is the change in fluorescence behavior induced by Ca²⁺ binding. Instead, in this study, we show that nanoscale secondary ion mass spectrometry (NanoSIMS) can be used as a tool to image the bound Ca²⁺ pool in the form of elemental Ca.

NanoSIMS is a high-resolution chemical imaging technique with a resolution of at least 200 nm, more than sufficient to examine intracellular Ca²⁺ in cells (Bonnin & Rizzoli, 2020). The technique works by using a negatively charged primary ion beam (O⁻) which strikes the surface of the sample and releases positively charged calcium ions. By using NanoSIMS, one can spatially map the distribution of Ca ions across a given area (Nuñez et al, 2017). The sample preparation methods for NanoSIMS, which involve embedding samples in plastic resin after several dehydration steps, serve to remove any free Ca²⁺ in the cells after fixation so that only bound Ca²⁺ ions make up the observed Ca signal.

Although NanoSIMS can image intracellular bound Ca²⁺ on its own, these images do not reveal directly new mechanistic information. To examine the role of bound Ca²⁺ in neuronal function, it is necessary to combine NanoSIMS analyses with other techniques, such as fluorescence microscopy (Vreja et al, 2015), which reveal the different functional compartments of the cell, enabling the correlation between cellular functionality and bound Ca²⁺. In this study, we therefore combined NanoSIMS analyses with fluorescent imaging of specific synaptic markers.

[1]Department of Neuro- and Sensory Physiology, University Medical Center Göttingen, Göttingen, Germany   [2]Excellence Cluster Multiscale Bioimaging (MBExC), Göttingen, Germany   [3]Center for Biostructural Imaging of Neurodegeneration (BIN), University Medical Center Göttingen, Göttingen, Germany   [4]Group of Computational Synaptic Physiology, Department of Neuro- and Sensory Physiology, University Medical Center Göttingen, Göttingen, Germany

Correspondence: elisa.bonnin@med.uni-goettingen.de; christian.tetzlaff@med.uni-goettingen.de; srizzol@gwdg.de

We specifically examined bound $Ca^{2+}$ in three locations: (i) the excitatory glutamatergic pre-synapse, (ii) the inhibitory GABAergic pre-synapse, and (iii) the excitatory (glutamatergic) post-synapse. Pre-synapses contain neurotransmitter-loaded synaptic vesicles, which release their contents onto synaptic receptors found in the post-synapse. When neurons are active, the vesicles from the excitatory synapses release glutamate, an amino acid that stimulates several types of ionotropic (ion channels) and metabotropic (non-channel) receptors in the respective post-synapse, thereby causing a depolarization and activation of the cell. The inhibitory vesicles contain GABA, a modified amino acid, which stimulates inhibitory receptors in its post-synapse, leading to a hyperpolarization (inactivation) of the cell (Martin, 2021). Both excitatory and inhibitory neurons are required in order to balance the flow of electrical information in the brain (Kajiwara et al, 2021).

Our results show that bound $Ca^{2+}$ is ubiquitous throughout the cell. However, its distribution is not homogenous throughout the cell. Bound $Ca^{2+}$ does not correlate to neuronal activity, but, surprisingly, it correlates negatively to synaptic release under spontaneous activity conditions.

# Results

### Relationship between bound $Ca^{2+}$ and cellular structure

To examine the relationship between bound $Ca^{2+}$ and synaptic function, we prepared samples of hippocampal neurons for correlative fluorescence imaging and NanoSIMS experiments. These samples were obtained using primary hippocampal cultures, following a classical protocol (Banker & Cowan, 1977) that dissociates the hippocampi of newborn rats. The resulting culture is a mixture of glia and neurons, with the majority of the neurons being glutamatergic (>90%) (Benson et al, 1994). Before fluorescent labeling, available coverslips were divided into two groups: one group to examine synaptic size and another to examine synaptic activity. Neurons from both groups were fixed and immunostained after 18 d in culture (DIV18), followed by plastic embedding and ultrathin sectioning.

Because the ions measured during a single NanoSIMS run are dependent on the source used (Nuñez et al, 2017; Bonnin & Rizzoli, 2020) and because $^{40}Ca$ does not always clearly show cell morphology, samples were first analyzed using the $Cs^+$ ion source to image $^{14}N$, $^{31}P$, and $^{32}S$, which represent the organic molecules composing the cells (e.g., proteins and nucleic acids). After this initial imaging step, the instrument was then switched to the $O^-$ ion source to image $^{40}Ca$ (Fig 1A). $^{39}K$ and $^{23}Na$ were also imaged using the $O^-$ ion source, as they are ubiquitous where cellular material is present (Fig S1A and B).

Comparing $^{40}Ca$ with $^{14}N$ and $^{32}S$ in cellular regions (defined by their high $^{14}N$ and $^{32}S$ signals) reveal two different groups, both positively correlated to $^{40}Ca$ (Fig 1B and C). One group, characterized by average $^{40}Ca$ counts >2 per ROI, makes up roughly 18.3% of all data points analyzed. A second group, characterized by average $^{40}Ca$ counts <2 per ROI, makes up the remaining ~81.6% of our samples. Although $^{40}Ca$ is correlated to $^{14}N$ and $^{32}S$ in both groups, this correlation has a much steeper slope in the former group than

in the latter. The existence of these two distinct groups of signals is expected, as $Ca^{2+}$-binding proteins are not thought to be present at equal levels in all cellular regions. Most cellular areas are basically devoid of $Ca^{2+}$, whereas a subset, presumably the areas containing $Ca^{2+}$-binding proteins, shows a strong correlation between this ion and the local protein amounts.

### No significant correlation between bound and free $Ca^{2+}$

To test whether a relationship exists between the bound and free $Ca^{2+}$ pools, we performed live imaging of the free $Ca^{2+}$ pool using Neuroburst (Sartorius), a genetically encoded calcium indicator. A separate batch of hippocampal neurons were transfected with the Neuroburst label at DIV10 and transferred to an imaging chamber at DIV17. The fluorescence intensity of the Neuroburst label was recorded for 2 min to capture spontaneous activity. After this, cells underwent field stimulation at 20 Hz for 3 s, followed by a recovery phase and renewed stimulation, for 30 s. Cells were then treated with a 5-μM ionomycin solution to measure the maximum possible fluorescence intensity, $F_{max}$ (Fig 2A).

The resulting movies show the normal neuronal activity of the cultures, in the form of bursts of $Ca^{2+}$ fluorescence. These represent high-frequency trains of action potentials that occur spontaneously at intervals of a few seconds. We recorded the following parameters from each observed neuron: peak intensity, or average peak amplitude during the 5-min recording; peak interval, or the frequency of subcomponent peaks; burst length, or the average width of burst events; burst interval, the frequency of bursts over the 5-min recording period; total intensity, or the summed amplitude of all peaks over the course of a single burst; maximum intensity normalized to initial intensity, and the average intensities of both stimulus peaks (Fig 2K).

Each imaging location was marked, and samples were embedded and processed for NanoSIMS analysis (Fig 2B and E). After NanoSIMS analysis, which was conducted in the same manner as in the previously described experiment (Fig 1), NanoSIMS images and live images were matched. The average intensity of the $^{40}Ca$ peak was recorded for each cell. The resulting bound $Ca^{2+}$ intensity was then compared with the free $Ca^{2+}$ parameters described above (Fig 2C–L). There did not appear to be any significant correlation between the amount of bound $Ca^{2+}$ and any parameter of the $Ca^{2+}$ fluorescence (Fig 2). This suggests that the free $Ca^{2+}$ fluorescence-derived measurements are not strongly correlated to the bound $Ca^{2+}$, at the whole-cell level.

### Bound $Ca^{2+}$ turnover is slow in both neuronal and nonneuronal cells

Having found that bound $Ca^{2+}$ does not correlate strongly to the dynamics of free $Ca^{2+}$, we wondered whether this lack of a relationship is due to bound $Ca^{2+}$ changing very rapidly. In principle, bound $Ca^{2+}$ could be a transient parameter. Neuronal activity, which takes place every few seconds in culture, results in abundant $Ca^{2+}$ entry into the cells, conceivably leading to gradual changes in the bound $Ca^{2+}$ pool. To tackle this issue, we decided to measure the bound $Ca^{2+}$ turnover, meaning the rate of exchange of these ions.

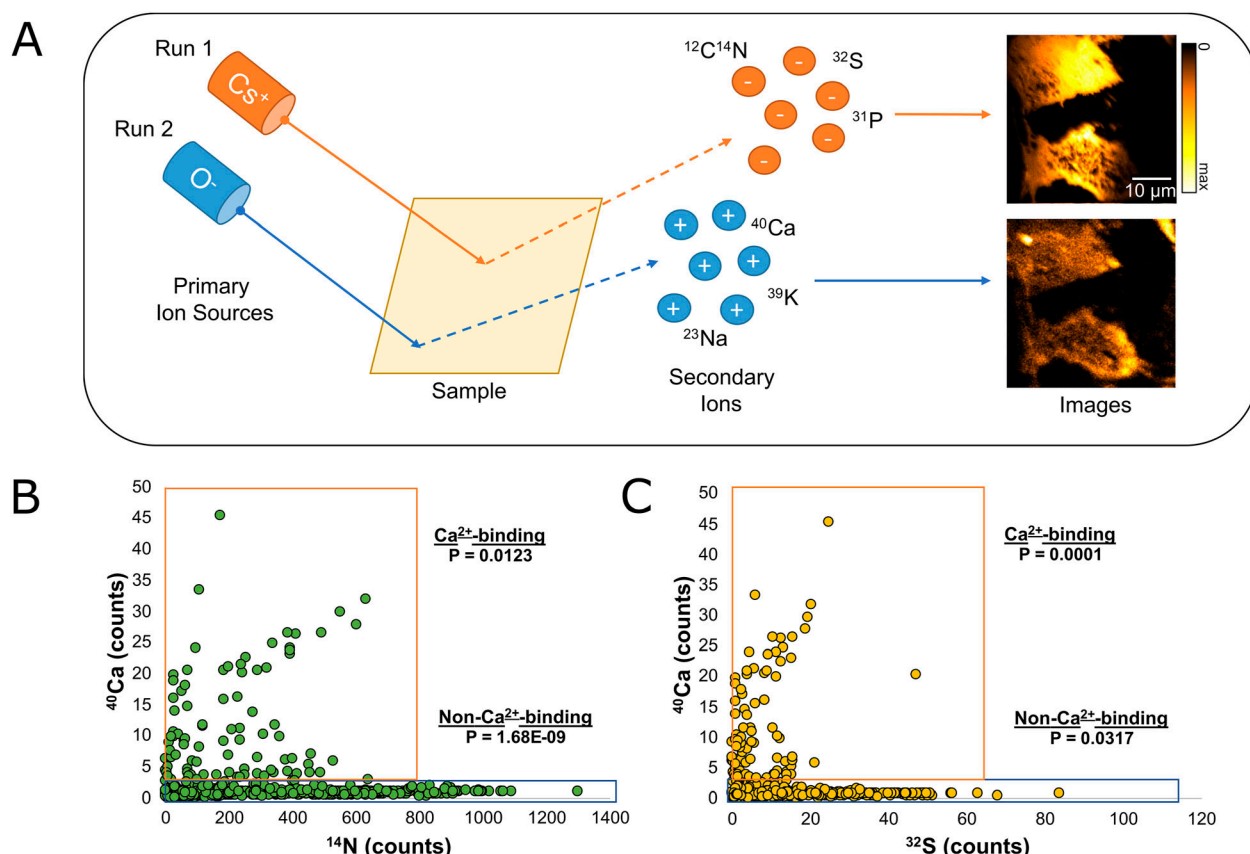

**Figure 1.  Comparison between calcium, sulfur, and nitrogen peaks as observed by nanoscale secondary ion mass spectrometry.**
**(A)** Nanoscale secondary ion mass spectrometry is a chemical-imaging technique which uses an ion source (Cs+ or O−) to produce a charged primary ion beam. When the beam strikes the surface of the sample, secondary ions of the opposite charge are created and measured. To analyze both $^{14}$N and $^{40}$Ca in our samples, it was necessary to first analyze the samples using the Cs+ source. Once all samples were analyzed using Cs+, the instrument was switched over to a radio-frequency RF source, and $^{40}$Ca was measured in a second run. **(B)** The intensity of the $^{40}$Ca peak at various points from each image is compared with the intensity of the $^{14}$N peak at the same location. The data are divided into two groups: a Ca$^{2+}$-binding group ($^{40}$Ca > 2, $N$ = 104, $P$ = 0.0123), and a non-Ca$^{2+}$-binding group ($^{40}$Ca < 2, $N$ = 462, $P$ = 1.68 × 10$^{-9}$). **(C)** The intensity of the $^{40}$Ca peak at various points from each image is compared with the intensity of the $^{32}$S peak at the same location. The data are divided into two groups: a Ca$^{2+}$-binding group ($^{40}$Ca > 2, $N$ = 104, $P$ = 0.0001), and a non-Ca$^{2+}$-binding group ($^{40}$Ca < 2, $N$ = 462, $P$ = 0.0317).
Source data are available for this figure.

We pulsed cells with isotopic calcium, $^{46}$Ca$^{2+}$, and measured the appearance of this isotope in the bound Ca$^{2+}$ pool. This process was very slow for fibroblasts (Human embryonic kidney-293 [HEK293] cells), only reaching, after 3 d of pulsing, ~4% of the maximum possible isotopic labeling (Fig S2A–C). This is remarkable, because the fibroblasts also divided during this time interval, and therefore needed to produce new calcium-binding proteins.

Ca$^{2+}$ turnover was also slow in neurons, reaching ~20% of the maximum labeling after 10 d of pulsing (Fig S3A–C). This implies that, in spite of rapid dynamics for free Ca$^{2+}$, the bound population is remarkably stable, both in neurons and in rapidly growing cells, as fibroblasts. This is all the more surprising, because the lifetimes of Ca$^{2+}$ buffer proteins as calbindin, calretinin, and parvalbumin are not particularly long (close to the median of all neuronal proteins, [Fornasiero et al, 2018]).

### The intensities of markers for inhibitory pre-synapses and excitatory post-synapses correlate to bound Ca$^{2+}$

To examine the relationship between bound Ca$^{2+}$ and pre- and post-synaptic parameters in neurons, we immunostained cultured hippocampal neurons for protein markers of the pre- and post-synapses. To analyze the pre-synapse, we used the vesicular glutamate transporter-1 (VGLUT1) and the vesicular GABA transporter (VGAT) as markers which are located in the pre-synapses of glutamatergic (excitatory) and GABAergic (inhibitory) neurons, respectively. As these markers label the pre-synaptic vesicle pool, their fluorescence intensity can be used to estimate the size of the respective synapses (Chaudhry et al, 1998; Vigneault et al, 2015). To image the post-synapse, we used post-synaptic density protein-95 (PSD95) as a marker of excitatory synapses (Ehrlich and Malinow, 2004). Immunostaining was conducted on fixed cells, and NanoSIMS analyses followed fluorescent imaging (Fig 3A and B, additional images in Fig S4). The same image regions were analyzed in the two techniques. As the axial resolution of the NanoSIMS instrument is around 10–20 nm in our experiments (Saka et al, 2014), this procedure does not lead to "mixed" measurements of multiple overlapping cells. In other words, when measuring from a region marked by neuronal or synaptic proteins, we are certain to provide ionic values from the respective neuronal cell. In total, ~60 neurons were imaged using these techniques. From these 60 neurons, 130

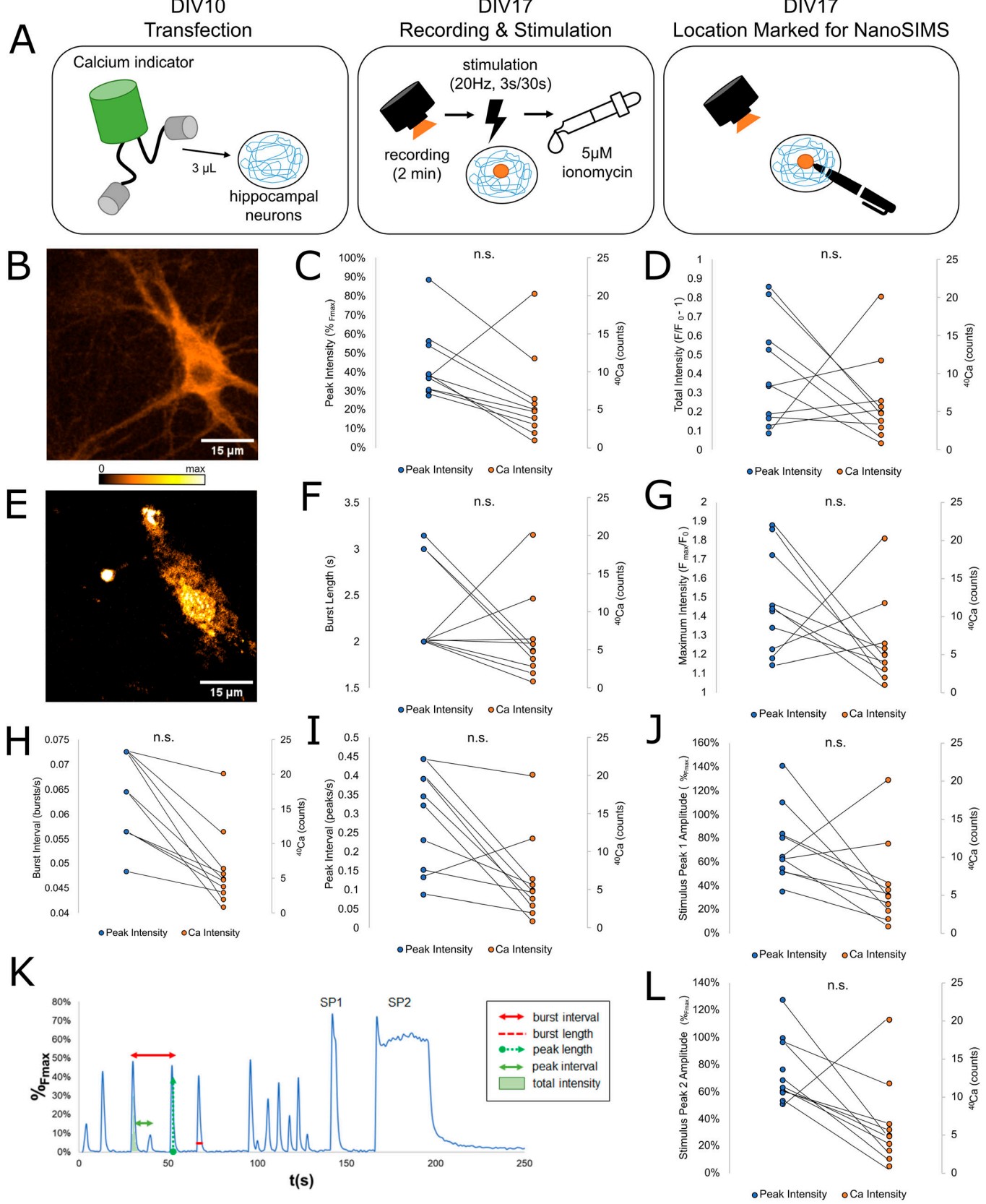

regions of interest were selected for VGLUT and $Ca^{2+}$ analysis (Fig 3C), 104 regions of interest were selected for PSD95 and $Ca^{2+}$ analysis (Fig 3D), and 126 regions of interest were selected for VGAT and $Ca^{2+}$ analysis (Fig 3E).

Comparing the intensities of these fluorescent markers in our samples with the intensity of the $^{40}Ca$ peak showed no significant correlation between bound $Ca^{2+}$ and the VGLUT intensity (Fig 3C). A positive correlation was present between PSD95 intensity and bound $Ca^{2+}$, albeit with a very limited $R^2$ value (Fig 3D). This correlation is likely artificially weakened by signal mixing, as the physical distance between the post-synapse and the glutamatergic pre-synapse is smaller than our image resolution, and the glutamatergic pre-synapse does not appear to correlate with bound $Ca^2$. The VGAT intensity is somewhat more strongly correlated to the amount of bound $Ca^{2+}$ (Fig 3E). This implies that the synapse size may correlate to the amount of Ca-buffering proteins, but only for the inhibitory pre-synapse and, to a lower degree, for the excitatory post-synapse. No significant correlations were found between these parameters and either $^{23}Na$ or $^{39}K$ (Fig S5A–C).

### Synaptic activity is inversely correlated to bound $Ca^{2+}$

The experiments presented above do not provide any functional parameters that can be correlated to the size of the bound $Ca^{2+}$ pool. To test this, we analyzed synaptic vesicle recycling. The behavior of synaptic vesicles was analyzed by conducting an initial 1-h immunostaining of live cells with antibodies targeting synaptotagmin 1 (Syt1), a vesicular calcium sensor (Kraszewski et al, 1995). During synaptic recycling, these antibodies are taken up by synaptic vesicles. This effectively labels any vesicle active during this 1-h time period (Sertel et al, 2021b), thereby measuring the size of the "recycling pool" of synaptic vesicles, which is a measure of the long-term activity capacity of the respective synapses (Truckenbrodt et al, 2018). After this staining was completed, cells were labeled for an additional 15 min with a fluorescent anti-mouse nanobody (NB), which identifies the Syt1 antibodies within the vesicles that are active during this shorter period, and thus provides a measure of the short-term activity of the respective synapses. The neurons were then fixed and further immunostained using a marker for synaptophysin (Syph), which labels the entire vesicle pool (Takamori et al, 2006) (Fig 4A). As with samples shown in Fig 3, cells were then embedded in resin and analyzed using both fluorescent microscopy and NanoSIMS (Fig 4B, additional images in Fig S6). In total, ~171 neurons were imaged using these techniques. From these neurons, 761 regions of interest were selected. The intensity of all three fluorescent markers were analyzed at each region of interest and compared with $Ca^{2+}$.

Comparisons between the intensity of these fluorescent markers and the intensity of the $^{40}Ca$ peak show that Syt1 labeling appears inversely correlated to bound $Ca^{2+}$, suggesting that synaptic activity decreases when bound $Ca^{2+}$ is high (Fig 4C). This result is initially surprising because Syt1 is a $Ca^{2+}$ sensor, which enables exocytosis in the presence of high $Ca^{2+}$ concentrations (Radhakrishnan et al, 2009; Jackman & Regehr, 2017). However, once free $Ca^{2+}$ becomes bound, it is no longer available to participate in biological reactions, implying that increasing the size of the bound $Ca^{2+}$ pool would decrease the amount of free $Ca^{2+}$ available for synaptic reactions, thereby reducing the long-term activity of the synapses.

We similarly observed a negative correlation between the amount of bound $Ca^{2+}$ at a given location and the intensity of the fluorescent nanobody (Fig 4D), suggesting that the short-term activity of the synapses is also regulated by bound $Ca^{2+}$. We did not observe any significant correlations between Syt1, the fluorescent nanobody and either $^{23}Na$ or $^{39}K$ (Fig S7A and B). Similarly, no correlation was observed between bound $Ca^{2+}$ and synaptophysin (Fig 4E), implying that these relationships are not reflective of the size of the vesicle pool. Some correlations could appear in this type of correlative imaging, simply because of biological material containing both proteins and metal ions. However, such correlations would be positive, because higher amounts of biological material would imply higher immunostaining levels and higher ion concentrations at the respective location (see Fig S7C). The negative correlations observed for Syt1 are difficult to explain by such effects.

Although the correlations between the Syt1 and NB labeling and bound $Ca^{2+}$ have low $R^2$ values (Fig 4C and D), they are highly significant statistically, and are visible even on small subsets of the analyzed data. Thus, although variation in bound $Ca^{2+}$ does not account for a substantial proportion of the dynamics of synaptic activity, it is nonetheless a factor that probably influences the synapse, as also explored in the following section.

### Computational model supports relation between size of bound $Ca^{2+}$ pools and synaptic activity

To investigate the inverse correlation between vesicle activity and bound $Ca^{2+}$, we devised a computational model of calcium binding and coupled it with the canonical model of short-term plasticity (Tsodyks et al, 1998) (Fig 5A and B). In this model, the normalized size of the

---

**Figure 2. Summary of correlated live imaging and NanoSIMS analyses of free and bound $Ca^{2+}$.**
**(A)** To observe the free $Ca^{2+}$ pool, we transfected cultured hippocampal neurons with a genetically encoded $Ca^{2+}$ indicator (Neuroburst) at DIV10. On DIV 17, we recorded the spontaneous activity of the neurons for 2 min, followed by stimulation at 20 Hz for 3 s, and a second stimulation at 20 Hz for 30 s. After the recording was completed, cells were treated with 400 $\mu$l of a 5 $\mu$M ionomycin solution in Tyrode buffer and imaged again to obtain $F_{max}$. After this, the location of the observed cell was marked on the coverslip for NanoSIMS analysis, and cells at that location were fixed and embedded. **(B, E)** Representative images of the same neuron observed using fluorescence microscopy (B) and NanoSIMS (E). **(C, D, F, G, H, I, J, L)** $Ca^{2+}$ intensity measured by NanoSIMS was compared to a variety of parameters of $Ca^{2+}$ activity: Comparison with (C) average peak fl. intensity (%$F_{max}$; $N$ = 10, $P$ = 0.5945) (D) average total fl. intensity ($F/F_0 - 1$; $N$ = 10, $P$ = 0.3132) (F) average burst length (s; $N$ = 10, $P$ = 0.6141) (G) maximum fl. intensity normalized to initial fl. intensity ($F_{max}/F_0$; $N$ = 10, $P$ = 0.0991) (H) average burst interval (bursts/s; $N$ = 10, $P$ = 0.1730) (I) average peak interval (peaks/s; $N$ = 10, $P$ = 0.5068) (J) the amplitude of the 3 s stimulus peak (Stimulus Peak 1, %$F_{max}$; $N$ = 10, $P$ = 0.8685), and (L) the amplitude of the 30 s stimulus peak (Stimulus Peak 2, %$F_{max}$; $N$ = 10, $P$ = 0.6547). **(K)** Representative image of the fluorescence intensity of a single cell during the entire recording period, with the free $Ca^{2+}$ parameters measured during this experiment labeled. Fluorescence intensities during the recording period were normalized as a percentage between initial intensity ($F_0$) and maximum intensity ($F_{max}$). Source data are available for this figure.

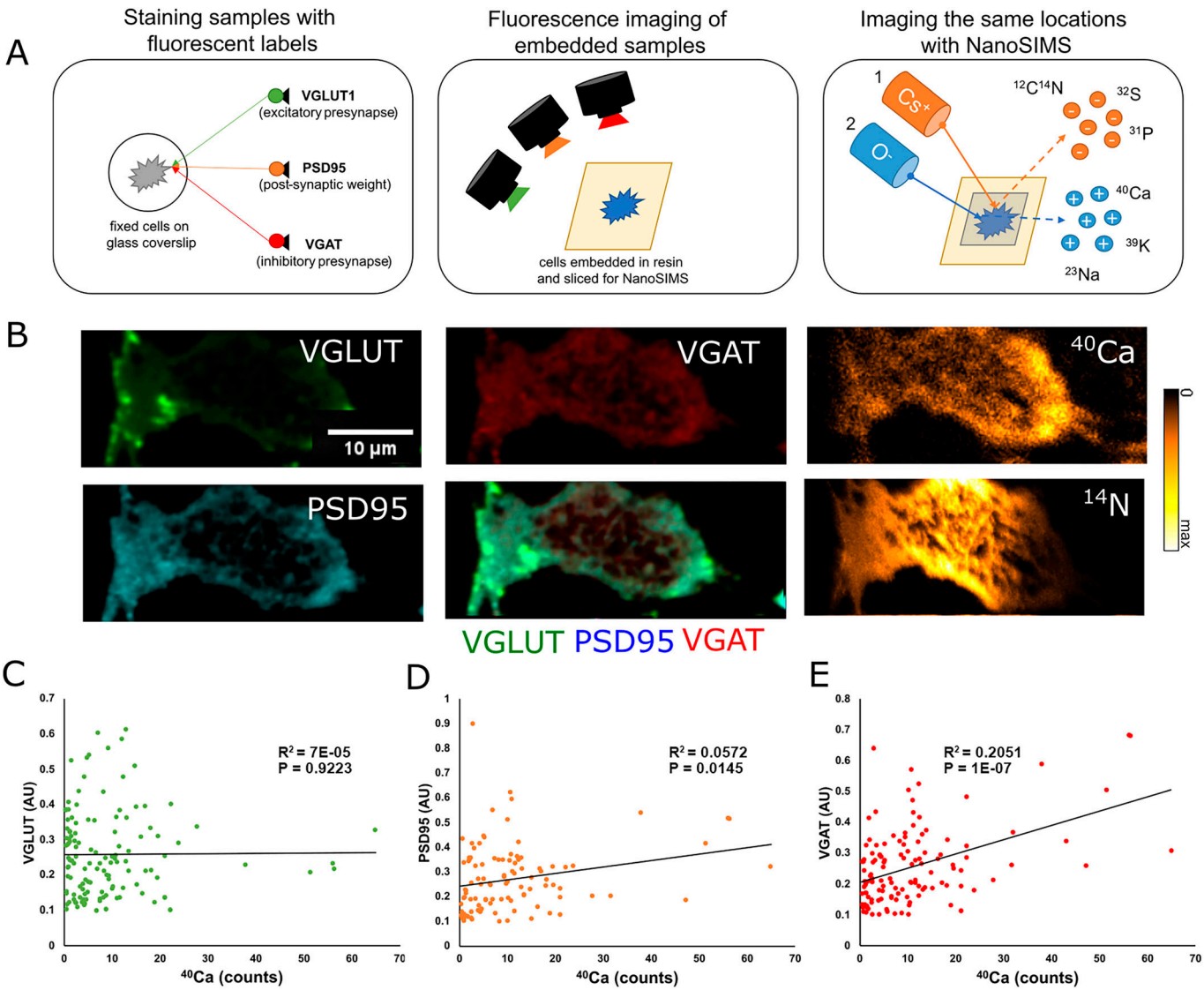

**Figure 3. Correlated fluorescence and NanoSIMS measurements of VGAT, VGLUT, and PSD95.**
**(A)** To compare synapse parameters with bound Ca²⁺, we stained fixed cells for VGLUT in the excitatory pre-synapse, PSD95 as a marker for the post-synaptic density, and VGAT in the inhibitory pre-synapse. Immediately after staining, samples were embedded for NanoSIMS analysis and cut into 200-nm-thin sections. Embedded samples were then imaged using a fluorescence microscope, and the same locations were analyzed using NanoSIMS. **(B)** Representative images of our three fluorescent parameters (VGLUT, PSD95, and VGAT) with a composite image (middle column, bottom). The images in the rightmost column are the result of NanoSIMS analyses of ⁴⁰Ca (upper right) and ¹⁴N (lower right). Fluorescent images and the ⁴⁰Ca image are cropped to show the same location as in the ¹⁴N image, which has the smallest raster size. **(C)** Graph comparing the fluorescence intensity of VGLUT to ⁴⁰Ca counts at selected regions of interest ($N$ = 130, $P$ = 0.9223). **(D)** Graph comparing the fluorescence intensity of PSD95 to ⁴⁰Ca counts at selected regions of interest ($N$ = 104, $P$ = 0.0145). **(E)** Graph comparing the fluorescence intensity of VGAT with ⁴⁰Ca counts at selected regions of interest ($N$ = 126, $P$ = 1 × 10⁻⁷).
Source data are available for this figure.

readily releasable pool of vesicles ($x$) and the release probability ($u$) are activity-dependent, as described by the following equations:

$$\frac{dx(t)}{dt} = \frac{1-x}{\tau_d} - uxR, \tag{1}$$

$$\frac{du(t)}{dt} = \frac{U-u}{\tau_f} + U(1-u)R, \tag{2}$$

$$w(t) = u(t)x(t). \tag{3}$$

Here, $R$ denotes the stimulation frequency, and $\tau_f$ and $\tau_d$ specify the synaptic facilitation and depression timescales, respectively. Equation (3) defines the time-dependent normalized synaptic weight. A schematic description is provided in Fig 5A (right). In the standard model of short-term plasticity (Tsodyks et al, 1998), the basal exocytosis probability $U$ enters as a free parameter that encapsulates the dependence of vesicle exocytosis rate on the intracellular calcium concentration. Thus, intracellular changes in free Ca²⁺ abundance because of, for example, transitions between bound and free

states modulate $U$. In extreme cases, lack and surplus of free $Ca^{2+}$ both saturate $U$ to 0 or 1, respectively. Therefore, we modeled the link between basal exocytosis probability of vesicles and free $Ca^{2+}$ by a sigmoid that reflects such plateaus:

$$U(c) = \frac{2}{1 + exp(-c/\lambda)} - 1. \qquad (4)$$

Here, $\lambda$ is a scaling factor and c denotes the abundance of intracellular free $Ca^{2+}$. Within the pre-synapse, free $Ca^{2+}$ can become bound, as described in the following transitional model:

$$\frac{dc(t)}{dt} = -\frac{c}{\tau_c} - \lambda_{cb}(B - b)c + \lambda_{bc}b + \mu R, \qquad (5)$$

$$\frac{db(t)}{dt} = -\frac{b}{\tau_b} + \lambda_{cb}(B - b)c - \lambda_{bc}b. \qquad (6)$$

In these equations, $b$ specifies the amount of bound calcium in the pre-synapse and $B$ describes the total binding capacity in the pre-synapse or buffer size. These equations read as follows: in general, both the amount of free and bound calcium decays, respectively, with timescales $\tau_c$ and $\tau_b$. Furthermore, free $Ca^{2+}$ can bind to $B–b$ available free buffer slots with a rate of $\lambda_{cb}$. Similarly, bound $Ca^{2+}$ may separate from buffer proteins, becoming free $Ca^{2+}$ proportional to a rate of $\lambda_{bc}$. In addition, the number of input spikes $R$ is assumed to linearly increase the level of free $Ca^{2+}$ by the rate $\mu$. Note that even though free $Ca^{2+}$ is rinsed away during NanoSIMS preparation, its interaction with bound $Ca^{2+}$ before fixation influences the size of the bound $Ca^{2+}$ pool and, thus, must be incorporated in the model, Fig 5A (left). A numerical solution of this coupled system for an arbitrary spike train is shown in Fig 5B.

To replicate the measurements used for Fig 4D, we analytically solved Equations (1), (2), (3), (4), (5), and (6) for the steady state condition with constant input rate R and computed the (stable) size of the readily releasable pool ($x$) for different levels of bound $Ca^{2+}$ $b$. We are interested in the size of readily releasable pool because it accounts for the immediate and short-term vesicle releases which were labeled by NB during immediate synaptic activity. Despite the multitude of parameters in the model, the overall decrease in the readily releasable vesicle pool size for an increasing number of bound $Ca^{2+}$ emerges for a wide range of parameters matching our experimental findings (Fig 5C).

In addition to matching experimental results, our model provides several predictions (Fig 5D). First, the model predicts a higher maximal synaptic weight $w_{max}$ for synapses with smaller buffer size. Second, our model yields the prediction that the maximum weight is reached by a higher input frequency $R_{max}$ for synapses with larger calcium buffer size than synapses with smaller size. Third, once all binding sites in the buffer are filled, synapses of any buffer size will behave similarly. As expected, this occurs at a high input rate regime, where continuous calcium injection and availability of free $Ca^{2+}$ promote binding even further.

## Discussion

By combining NanoSIMS analyses with fluorescent imaging, we have demonstrated that we can image the bound $Ca^{2+}$ pool in neuronal cells and relate the size of this pool to functional parameters within the synapses (Fig 1). Using this approach, we could determine that free and bound $Ca^{2+}$ did not correlate at the whole-cell level (Fig 2). However, we could determine that bound $Ca^{2+}$ does correlate significantly with several parameters at the synapse level. The size of the bound $Ca^{2+}$ pool is positively correlated to the levels of PSD95 and VGAT (Fig 3) and is negatively correlated to synaptic activity (Fig 4). Extending a well-established model of short-term synaptic plasticity by calcium dynamics, we show that the negative correlation between synaptic activity and bound $Ca^{2+}$ holds for a wide range of parameters. In addition, our model leads to predictions about the influence of the size of the calcium buffer on input-dependent vesicle release.

In our results, we have not observed any correlation between free $Ca^{2+}$ and bound $Ca^{2+}$ at the level of the whole cell (Fig 2), although in the synapse, bound $Ca^{2+}$ seems to be anti-correlated to synaptic activity, implying an anti-correlation to free $Ca^{2+}$ concentrations. This discrepancy is probably because of differences between the dynamics of $Ca^{2+}$ regulation in different cellular compartments. $Ca^{2+}$ regulation is extremely complex (Berridge, 1998; Pivovarova & Andrews, 2010; Jackman & Regehr, 2017; Mammucari et al, 2018), and while our methods can examine the relative size of the bound $Ca^{2+}$ pool, NanoSIMS analyses alone cannot determine to which proteins or compounds the observed $Ca^{2+}$ is bound. NanoSIMS is a type of dynamic SIMS, meaning that the instrument uses a primary beam powerful enough to break apart chemical bonds, so that only elemental ions and small fragments (e.g., $CN^-$) are measured. This configuration produces a high yield of secondary ions, allowing NanoSIMS to image the spatial distribution of these ions with high resolution at the expense of information at the molecular level (Nuñez et al, 2017). Therefore, it is probable that clearer results can be obtained in synapses than in the very complex cell body. Given the variety of proteins which bind $Ca^{2+}$ within cells, it is likely that whole-cell measurements cannot fully capture the complexity and diversity of calcium dynamics. At the same time, the bound $Ca^{2+}$ population in the cell body does not appear to be very dynamic because it exchanges very slowly with the extracellular $Ca^{2+}$ (Figs S2 and S3).

At the synaptic level, a correlation of bound $Ca^{2+}$ to VGAT is expected in the context of previous research, which shows more $Ca^{2+}$ buffering in inhibitory synapses because of the high abundance of $Ca^{2+}$ buffering proteins such as parvalbumin (Lee et al, 2000; Verderio et al, 2004; Hu et al, 2014; Pelkey et al, 2017; Courtney et al, 2018). Similarly, previous research has suggested the presence of sizeable amounts of Ca-buffering proteins in the post-synapse (Augustine et al, 2003; Giese, 2021), supporting our observation that bound $Ca^{2+}$ is correlated to PSD95 (Fig 3D). Importantly, some of the differences noted here between different types of synapses may be because of the presence of mitochondria at higher levels, and with higher frequencies, in particular synaptic compartments (Rossi & Pekkurnaz, 2019). The mitochondrial presence would strongly influence local $Ca^{2+}$ buffering, leading to different organizations in the respective $Ca^{2+}$-binding machineries.

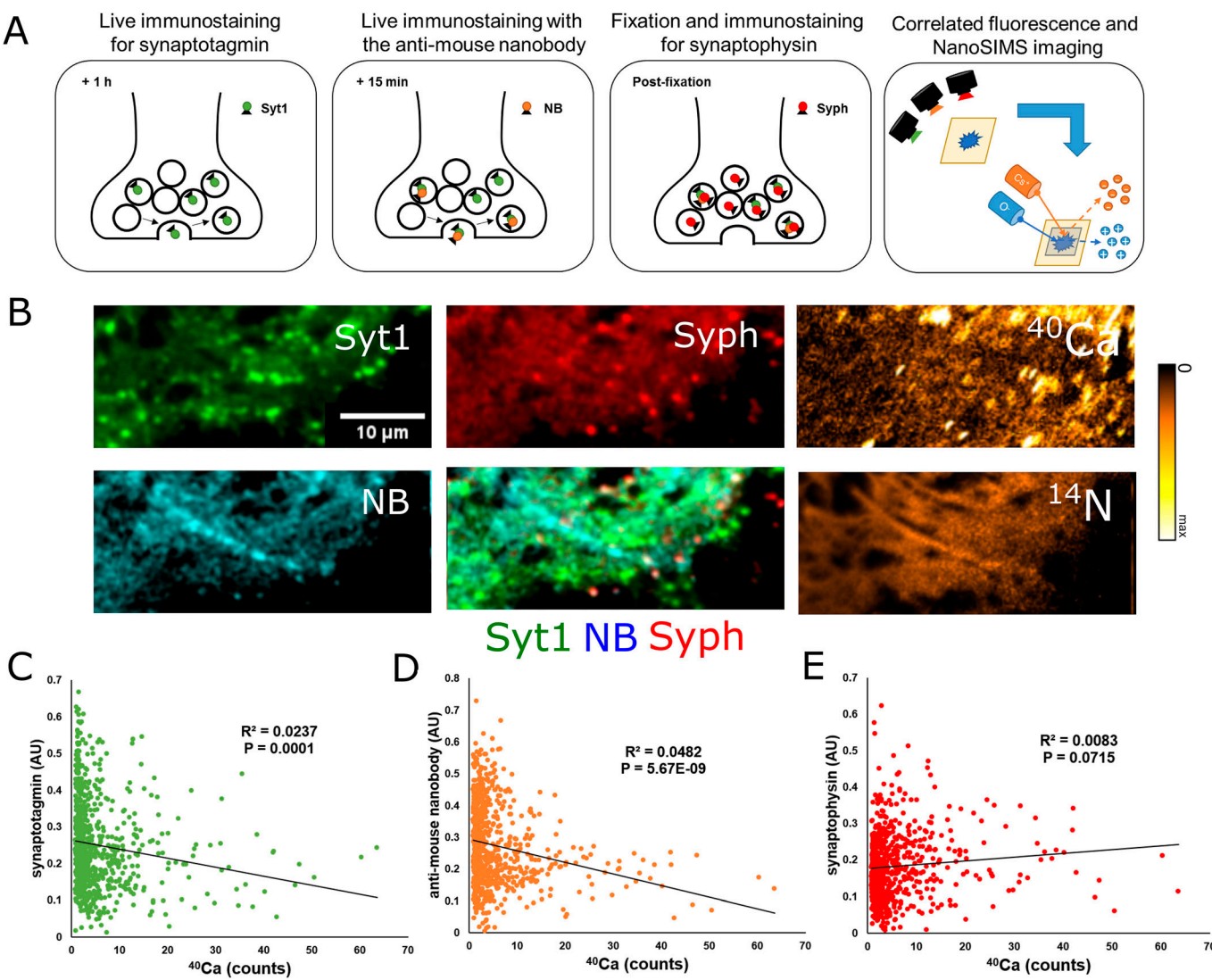

**Figure 4. Correlated fluorescence and NanoSIMS measurements of synaptotagmin 1 (Syt1), the anti-mouse nanobody (NB), and synaptophysin (Syph).**
**(A)** To compare synaptic activity with bound Ca²⁺, we stained live cultured hippocampal neurons with a marker for synaptotagmin 1 for 1 h to label the entire active vesicle pool. After this, we stained the same culture live for 15 min with a secondary nanobody to sample immediate activity. Cells were then fixed and immunostained with Syph to label the entire vesicle pool. Immediately after staining, samples were embedded for NanoSIMS analysis and cut into 200 nm thin sections. Embedded samples were then imaged using a fluorescence microscope, and the same locations were analyzed using NanoSIMS. **(B)** Representative images of our three fluorescent parameters (Syt1, NB, Syph) with a composite image (middle column, bottom). The images in the rightmost column are the result of NanoSIMS analyses of ⁴⁰Ca (upper right) and ¹⁴N (lower right). Fluorescent images and the ⁴⁰Ca image are cropped to show the same location as in the ¹⁴N image, which has the smallest raster size. **(C)** Graph comparing the fluorescence intensity of Syt1 to ⁴⁰Ca counts at selected regions of interest ($N = 761$, $P = 0.0001$). **(D)** Graph comparing the fluorescence intensity of the anti-mouse nanobody to ⁴⁰Ca counts at selected regions of interest ($N = 761$, $P = 5.67 \times 10^{-9}$). **(E)** Graph comparing the fluorescence intensity of Syph to ⁴⁰Ca counts at selected regions of interest ($N = 761$, $P = 0.0715$).
Source data are available for this figure.

We further observed that synaptic activity on time scales from 15 min to 1 h appears to decrease when the bound Ca²⁺ pool is large, suggesting that, under spontaneous activity conditions, bound Ca²⁺ inhibits synaptic release. This finding is initially surprising because of synaptotagmin's role in regulating free Ca²⁺ during synaptic release (Rama et al, 2015; De Jong et al, 2016; Jackman & Regehr, 2017). However, during synaptic transmission, free Ca²⁺ penetrates from the outside of the synapses and is then buffered (Collin et al, 2005). Synapses that have a larger buffering capacity (high amounts of Ca²⁺-binding proteins) would

induce a decrease of Ca²⁺ within the synapses more rapidly than in synapses with lower amounts of buffer proteins, resulting in lower activity (Vyleta & Jonas, 2014; Mironov, 2019), which would explain our observations.

We tested these observations using a numerical model, which supports the measured negative correlation between vesicle recycling activity and bound Ca²⁺. This correlation was not sensitive to model parameters. Our analysis also affirms the expectation that bound Ca²⁺ regulates synaptic activity as long as the calcium buffer is not full. In the case of an entirely occupied buffer, all calcium

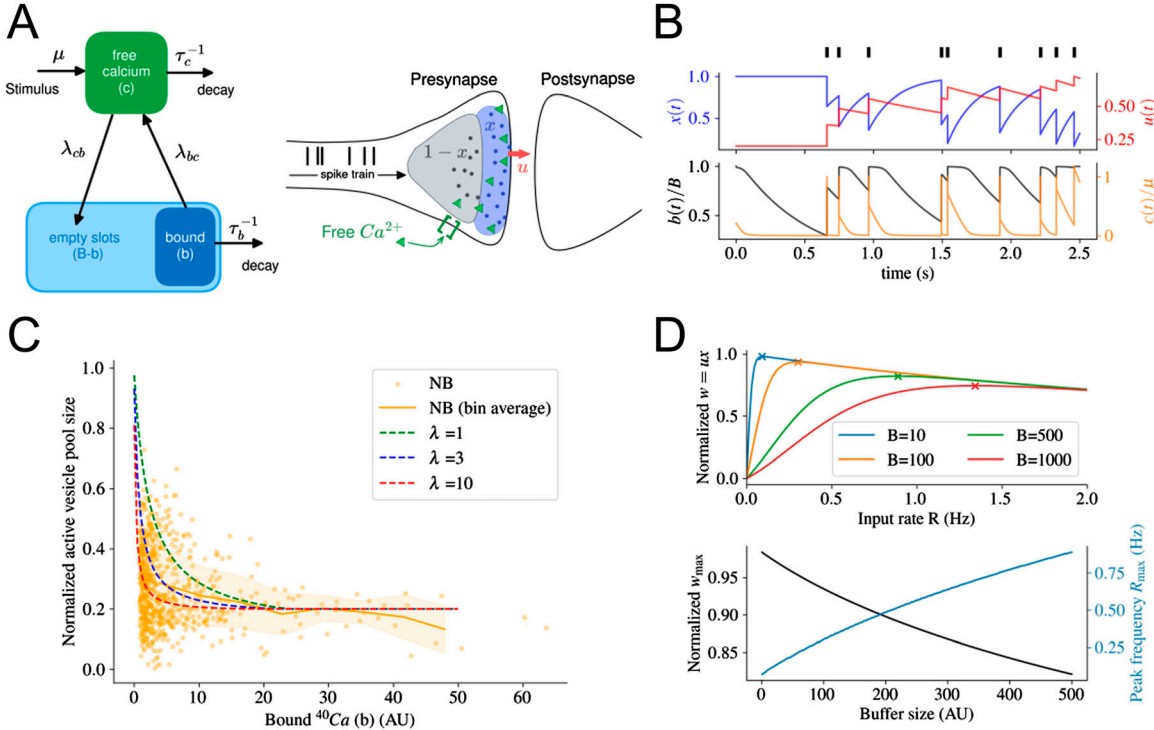

**Figure 5. Summary of computational model and its prediction.**
**(A)** (left) Schematic of the transition model of calcium from the bound state to free, and vice versa. Transition rates are denoted on top of each arrow. Note that transition from free to bound $Ca^{2+}$ is possible only if the buffer is not fully occupied. (Right) Short-term synaptic plasticity originates from the calcium-dependent release of vesicles in the pre-synapse (Tsodyks et al, 1998). Of those vesicles that participate in exocytosis (active pool), the fraction $x(t)$ (blue shade) is ready to release, and with a probability $u(t)$ will be fused to the cell membrane if an action potential arrives. The release probability $u(t)$ is regulated by the concentration of free calcium (green triangles) in the pre-synapse. However, in the pre-synapse, free calcium can also be absorbed to become bound calcium (not shown). (Right) **(B)** Temporal evolution of synaptic variables ($u, x$ - top panel) and bound and free ($b, c$ - bottom panel) calcium for a given input spike train (black bars on the top). In our model, we substituted the incoming spike train by a constant input of rate $R$. **(C)** The steady-state size of the readily releasable vesicle pool ($x$) decays as the amount of bound $Ca^{2+}$ ($b$) increases for a wide range of parameter $\lambda$. The overlaid scattered data are the intensity of anti-mouse NB fluorescence, same as in Fig 4D. The solid orange line and the range of dashed areas show the mean and one SD in each bin. **(D)** (Top) In steady state, the normalized synaptic weight depends on the input frequency and presynaptic calcium buffer size $B$. Crosses mark the maximum synaptic weight. (Bottom) Maximum synaptic weight and corresponding input frequency are related to the presynaptic buffer size.

surplus inevitably becomes free, decoupling synaptic activity from bound states.

Another prediction of our computational model is the inverse correlation between the maximal synaptic weight and the buffer size. Given the assumption that larger synapses have a larger calcium buffer, our model indicates that the resulting presynaptic calcium dynamics lead to the fact that larger synapses use a smaller fraction of available vesicles. In addition, this fraction is being mobilized by a higher input frequency.

One important limitation of our experimental method is that, whereas it compares fluorescence intensities at one location with the intensity of the $^{40}Ca$ peak observed using NanoSIMS at the same location, it cannot compare the size of the bound $Ca^{2+}$ pool between different cellular proteins. Doing so is a possible avenue for future research; however, we note that, although the NanoSIMS produces good estimates of the relative size of the $Ca^{2+}$ pool, obtaining more quantitative information (such as the exact concentration of $Ca^{2+}$ within the cell) would require the use of calibrated standards, the production of which is nontrivial (Gabitov et al, 2013).

Similarly, it is important to note that our study analyzes the bound $Ca^{2+}$ pool after chemical fixation and plastic embedding, implying that we are only measuring $Ca^{2+}$ bound to structures that can withstand these invasive steps. At the moment, we have no way of quantifying how much calcium is lost during sample preparation. Some compounds that are known to be calcium-binding, including lipids (Bagur & Hajnóczky, 2017), are removed during embedding. To obtain a complete picture of intracellular-bound $Ca^{2+}$, it will be necessary to measure the sample without embedding, by adapting a cryo-stage for use in NanoSIMS. Efforts are underway to construct a reliable cryo-NanoSIMS, and the availability of such a device is expected to increase in future (Jensen et al, 2016; Meibom et al, 2023). This represents a viable path for future study.

These limitations aside, however, our results suggest that the bound $Ca^{2+}$ pool is not merely inert, but rather is an important regulator of synaptic activity. This opens a new field for SIMS investigations correlating physiology with bound ionic concentrations, something that is not within the range of possibility for fluorescence measurements alone.

## Materials and Methods

### Experimental design

The goal of the study was to examine the relationship of bound $Ca^{2+}$ to synaptic function in neurons using a combination of NanoSIMS and fluorescence imaging. Because all free $Ca^{2+}$ is washed out after the embedding procedure outlined below, all $Ca^{2+}$ imaged using NanoSIMS belongs to the bound $Ca^{2+}$ pool. Cultured cells were divided into two groups for correlated NanoSIMS and fluorescence analyses. The first group, intended to examine synaptic size, was immunostained for the glutamatergic pre-synaptic protein VGLUT1, the GABAergic pre-synaptic protein VGAT, and the post-synaptic protein PSD95. The second group was set aside to examine synaptic activity and was immunostained live for synaptotagmin to label the entire active vesicle pool, and for a secondary fluorescent nanobody to sample immediate activity. After fixation, these samples were immunostained for synaptophysin to label the entire vesicle pool.

### Hippocampal cultures

Rats (*Rattus norvegicus*, wild type, Wistar) for the primary neuron cultures were handled according to the specifications of the University of Göttingen and of the local authority, the State of Lower Saxony (Landesamt für Verbraucherschutz, LAVES, Braunschweig, Germany). All experiments and procedures were approved by the local authority, the Lower Saxony State Office for Consumer Protection and Food Safety (Niedersächsisches Landesamt für Verbraucherschutz und Lebensmittelsicherheit). All methods are reported in accordance with ARRIVE guidelines and were carried out in accordance with relevant guidelines and regulations. The experimental procedure was approved by the relevant institutional entity, the Tierschutzbüro of the University Medical Center Göttingen (approval number T 09/08).

Neuronal hippocampal cultures were obtained by collecting the dissociated hippocampi of newborn rat pups, as described in previous studies (Banker & Cowan, 1977; Kaech & Banker, 2006; Truckenbrodt et al, 2018; Yousefi et al, 2021). In summary, brains from rat pups were extracted on post-natal day 2, and the hippocampi were separated. Tissue debris was removed from dissected hippocampi by washing with HBSS (Invitrogen). After this wash, the hippocampi were incubated for 1 h in a sterile-filtered enzyme solution made up of 10 ml DMEM (Thermo Fisher Scientific), 2 mg cysteine, 100 mM $CaCl_2$, 50 mM EDTA, and 25 U papain, equilibrated with carbogen for 10 min. Hippocampi were then washed with HBSS and incubated for 15 min in an inactivating solution made up of 2 mg albumin and 2 mg trypsin inhibitor in DMEM containing 10 ml FCS.

Coverslips were prepared by treatment with nitric acid, thorough washing with $ddH_2O$, sterilization and overnight coating with 1 mg/ml poly-l-lysine. Coverslips were then washed once more with $ddH_2O$ before being incubated with a solution of MEM (Thermo Fisher Scientific) supplemented with 10% horse serum, 3.3 mM glucose, and 2 mM glutamine. Neurons from dissected hippocampi were then plated at a concentration of ~30,000/$cm^2$. Cells were left

to adhere for 1–4 h at 37°C in a cell incubator, with 5% $CO_2$. After adhesion, cellular medium was replaced with Neurobasal-A medium (Gibco, Life Technologies), containing 1:50 B27 supplement (Gibco) and 1:100 GlutaMAX (Gibco). 5-fluoro-2′-deoxyuridine was added to the cells after 2–3 d in culture to avoid glial proliferation.

Neurons were kept in culture at 5% $CO_2$ and 37°C for 18 d before fluorescent labeling. On day in vitro (DIV) 10, the calcium concentration of the culture medium was increased by 10 mM through the addition of dissolved calcium carbonate ($CaCO_3$). Cells were then returned to the incubator and allowed to incubate for the remaining 8 d.

### Immunostaining

Prior to fixation (DIV18), neurons were divided into two groups, one group to be labeled for VGAT, PSD95, and VGLUT simultaneously, and one group to be labeled for synaptotagmin, synaptophysin, and the fluorescent nanobody. Cells in the synaptotagmin group were immunostained live for synaptotagmin with a 1:500 dilution of the primary antibody (synaptotagmin, Synaptic Systems, Catalog #105 311 C2) and incubated for 1 h. After incubation, cells were washed three times with Tyrode buffer (124 mM NaCl, 5 mM KCl, 2 mM $CaCl_2$, 1 mM $MgCl_2$, 30 mM D-glucose, and 25 mM HEPES) and incubated in medium labeled with a 1:500 dilution of the anti-mouse nanobody (FluoTagx2 anti-mouse with Atto542, Catalog #N1202-At54; NanoTag) for 15 min.

After live staining, all neurons from all groups were washed three times in Tyrode buffer, and were fixed for 30 min at room temperature with 4% PFA (Sigma-Aldrich) as performed in previous studies (Sertel et al, 2021a). Samples were quenched by a 30-min incubation at room temperature with 100 mM $NH_4Cl$ in PBS. Cells were then washed three times with a permeabilization solution (3% BSA, 1:10,000 Triton-X-100 in PBS). This was followed by another 1 h incubation step with primary antibodies (Table S1) in the permeabilization solution. Samples were once more washed three times in permeabilization solution and then incubated for 1 h with secondary antibodies (Table S1), also in the permeabilization solution. Samples were stored in PBS overnight at 4°C before embedding.

### Sample preparation for NanoSIMS

For correlated NanoSIMS and fluorescence imaging, samples were embedded in LR white resin (London Resin Company) as described in a previous study (Jähne et al, 2021). Samples were first partially dehydrated through a 5 min incubation step with 30% ethanol (in $ddH_2O$), followed by three 5 min incubation steps in 50% ethanol (in $ddH_2O$) with rotation at 150 rpm using a Rotamax 120 platform shaker (Heidolph Instruments). After this, samples were incubated for 1 h in a 1:1 solution of pure LR white and 50% ethanol (in $ddH_2O$). The coverslips were then transferred to a new plate to avoid sticking and incubated in pure LR white for 1 h at room temperature. After incubation, coverslips were removed from the plate. Beem capsules (BEEM Inc.) were placed on top of the coverslips. One drop of LR white accelerator (London Resin Company) was mixed with 10 ml LR white. This mixture was then added to the capsules to create a seal. After hardening for 30 min, capsules were topped with

a fresh mixture of LR white and LR white accelerator. Samples were then cured at 60°C until the LR white resin had fully hardened (at least 90 min). After cooling, the coverslips were removed from the capsules. An ultramicrotome (EM UC6; Leica Microsystems) was used to cut 200-nm-thin sections from the samples, which were placed onto silicon (Si) wafers (Siegert Wafer GmbH) for analysis.

## Fluorescence imaging

For fluorescence imaging, Si wafers were mounted onto glass slides and imaged at 60x using an inverted epifluorescence microscope (Nikon Eclipse Ti-E; Nikon). After imaging, oil was carefully removed from samples by repeated additions of isopropanol, which were subsequently removed from the sample using compressed air. As a final cleaning step, a small amount of a chloroform and iso-propanol solution (~1:20) was added to the sample surface. After the oil was removed, samples were transferred to the NanoSIMS for further analysis.

## NanoSIMS analysis

SIMS images were obtained using a NanoSIMS 50L instrument (Cameca). Embedded samples were initially imaged using a Cs+ positive ion source to capture cell morphology. Secondary ions were generated using a primary current of ~30 nA (primary aperture D1 = 2). Before each measurement, Cs+ ions were implanted at high current (L1 = 24,000, D1 = 1) on an area larger than the raster size used for analysis until steady-state conditions were achieved. Images were then taken at a raster size of 40 × 40 $\mu$m, and a resolution of 256 × 256 pixels, with 5,000 cts/px. The estimated resolution from these images is ~156.25 nm/px. To minimize sample degradation, one image was collected with these settings at each location. The following peaks were collected for each run: $^{12}C^{14}N$ (referred to as $^{14}N$ in this report), $^{31}P$, and $^{32}S$.

After images were obtained from all samples using the Cs+ source, the instrument was switched to a radio frequency (RF) O⁻ source for calcium measurements. Secondary ions were generated using a primary current of ~200 nA (primary apertures D1 = 2, D0 = 2). Similar to the images in the paragraph above, O⁻ ions were implanted at high current (L1 = 24,000, D1 = 1) on an area larger than the raster size. The same locations as were imaged using the Cs+ source were imaged using the O⁻ source at a raster size of 50 × 50 $\mu$m, and a resolution of 256 × 256 pixels, with 5,000 cts/px. The estimated resolution from these images is ~195.31 nm/px. A larger raster size was used to ensure that the entire area imaged with Cs+ was visible in the O⁻ images. Three images were obtained at each location with the settings above. The following masses were collected during each run: $^{23}Na$, $^{39}K$, $^{40}Ca$, and $^{48}Ca$. Each image shown in this article is a result of the summation of all three image layers taken during analysis.

## Calcium imaging

Neuroburst (Sartorius) was used as a genetically encoded calcium indicator, following a method used in a previous study (Sertel et al, 2021a). We used the Incucyte Neuroburst Orange Lentivirus, which is a third-generation, based on HIV, VSV-G pseudotyped lentivirus

particle, provided in a suspension form, in DMEM. Virus transduction was performed on DIV10, with 3 $\mu$l of Neuroburst added to each well plate. The medium was replaced every few days, removing one third of the neuronal medium and adding a similar amount of fresh medium to maintain the cultures at a volume of ~1 ml. On DIV17, coverslips were placed in Tyrode buffer and imaged at 37°C using the same Nikon Eclipse Ti-E inverted epifluorescence microscope as used for fluorescence imaging. Coverslips were imaged for 2 min in 1 s intervals (1 Hz). After 2 min of imaging, neurons underwent field stimulation for 3 s at 20 Hz, and then for 30 s at 20 Hz. After neurons resumed spontaneous activity, the Tyrode buffer in the imaging chamber was replaced with 400 $\mu$l of a 5 $\mu$M solution of ionomycin (Sigma-Aldrich) in Tyrode. Cells were allowed to acclimatize to the ionomycin addition, after which cells were imaged for 15 s in 1 s intervals.

The location of the imaged cells on the coverslip was marked after each recording. After recordings were completed, cells were fixed and embedded according to the sample preparation procedure described earlier in this article, with the exception that the coverslips were not removed from the plate between the 1 h incubation step with LR white and 50% ethanol and the 1 h incubation step with pure LR white. The Beem capsule was placed over the marked location of the cell, and coverslips were not moved during the hardening and sealing steps, to ensure that the same cells as imaged were embedded. NanoSIMS analysis proceeded as previously described. Cells were chosen by comparing the arrangement of cells visible in NanoSIMS with those of cells imaged during live imaging.

## Ca$^{2+}$ turnover in HEK293 cells and neurons

HEK293 cells were maintained in culture in DMEM with 5% FCS, 2 mM L-glutamine, 60 U/ml penicillin, and 60 U/ml streptomycin. In preparation for the experiment, cells were seeded into a six-well plate containing coverslips coated with poly-l-lysine. Five out of six wells contained DMEM Ca-free medium (Thermo Fisher Scientific) with 5% FCS, 2 mM L-glutamine, 60 U/ml penicillin, and 60 U/ml streptomycin. The calcium concentration of the medium was increased to 2 mM by the addition of a solution of calcium carbonate enriched in $^{46}Ca$ (5% $^{46}CaCO_3$, 95% $^{40}CaCO_3$; Cambridge Isotope Laboratories). The final well contained normal DMEM and was used as a control experiment.

Coverslips remained in culture for a set amount of experimental time, namely 1 and 6 h, 1 or 3 d. Control cells remained in culture for 3 d. After each experimental period was completed, cells were fixed for 30 min at room temperature with 4% PFA (Sigma-Aldrich) and embedded in LR white medium as previously described in the article.

Hippocampal neurons were prepared in the same way as all other neuronal cultures used here. On DIV 4, $^{46}Ca$-enriched calcium carbonate solution was added to each well-plate, raising the Ca$^{2+}$ concentration of the buffer up to ~16 mM. Half of the coverslips were fixed and prepared for NanoSIMS on DIV8, after 4 d in the $^{46}Ca$ experiment. The remaining coverslips were fixed and prepared for NanoSIMS on DIV14 after 10 d in the $^{46}Ca$ experiment.

NanoSIMS analyses used a raster size of 70 $\mu$m (~273 nm/px) for all images made using HEK293 cells, and a raster size of 20 $\mu$m

(~78 nm/px) for all images made using hippocampal neurons labeled with $^{46}$Ca. Analyses otherwise proceeded using the same method as for other samples used in this study, with the additional collection of the $^{46}$Ca peak using the radio frequency (RF) O$^-$ source. After analyses, mean $^{46}$Ca/$^{40}$Ca ratios of each cell were collected by averaging the intensity of the $^{46}$Ca peak across the entire area of the cell and dividing that average by the average intensity of $^{40}$Ca across the same area.

In all experiments, the $^{46}$Ca/$^{40}$Ca ratio of the controls is slightly higher than expected for the natural abundance (~0.0413‰) of this isotope. This is expected, because of counting statistics, as the NanoSIMS instrument collects data in units of counts (Fitzsimons et al, 2000). Because the natural abundance of $^{46}$Ca is so low (0.004% of all Ca atoms), the collection of a single $^{46}$Ca count in an image can artificially increase the resulting $^{46}$Ca/$^{40}$Ca ratio.

### Image processing & correlation

NanoSIMS images were initially processed using a custom Matlab script (Supplemental Data 1) (the Mathworks Inc.) adapted from the analysis script used in previous studies (Gagnon et al, 2012; Bonnin et al, 2021). Image layers were aligned relative to the second image from each element before summation. The resulting matrices were then saved as text files, and images were generated using FIJI/Image J (NIH, Bethesda, MD, USA).

Fluorescence images were extracted using the Bio-Formats plug-in in FIJI/ImageJ. Fluorescence images and $^{40}$Ca SIMS images were then aligned by hand. A second Matlab processing script was used to select ROIs from the aligned images (Fig S8A–C). ROIs were selected by hand based on high signal in fluorescence. For comparison between $^{14}$N, $^{32}$S, and $^{40}$Ca (Fig 1), ROIs were selected based on high signal in either $^{14}$N or $^{32}$S. For the VGAT/PSD95/VGLUT1 image set (Fig 3), ROIs were selected based on high signal either in VGAT or in VGLUT, denoting either an inhibitory or excitatory synapse, respectively, or based on high signal in PSD95, denoting the location of the excitatory post-synapse. For the Syt1/NB/Syph image set (Fig 4), ROIs were selected based on high intensity in any of the three fluorescent markers. Locations that did not correspond to cellular material were excluded from analysis.

The ROI selection script works by asking the user to select background points at each image. Once background levels are recorded, the user can select ROIs. The script then records the corresponding intensity value at each selected location for each associated image, subtracted from the background. ROIs are defined as circles with a radius of 6 pixels, leading to an interior area of ~113 pixels. Intensity values for each pixel within an ROI are averaged, so that a single average value per ROI is recorded. Before further analysis, data points with $^{40}$Ca intensities at or below background level were removed. This was done by assuming a normal distribution of background $^{40}$Ca intensities with a mean of 0 and filtering out points within one SD of the mean.

### Statistical analysis

The significance of the correlations between bound Ca$^{2+}$ and either $^{14}$N and $^{32}$S (Fig 1) or fluorescence intensities (Figs 3, 4, S1, S5, and S7)

was determined using a linear regression test, with N and P-values as reported in the associated figures. A Bonferroni multiple comparison correction was applied to the data displayed in Figs 4, S1, S5, and S7, to correct for multiple parameters being compared against each other. For the free Ca$^{2+}$ imaging experiment, significance between free Ca$^{2+}$ parameters and bound Ca$^{2+}$ was determined using a two-tailed Pearson correlation test, with N and P-values as reported in the associated figure (Fig 2). For Ca$^{2+}$ turnover experiments (Figs S2 and S3), experimental groups were compared against each other using the Kruskal–Wallis one-way analysis of variance to determine whether statistically significant differences existed within the groups. After this, Dunn's test for multiple comparisons was used to perform pairwise comparisons between each independent group.

### Analytical solution of computational model

The steady-state solution of the dynamical system discussed in the main text can be found by setting all time derivatives to zero.

$$\frac{dx(t)}{dt} = \frac{1-x}{\tau_d} - uxR = 0 \tag{7}$$

$$\frac{du(t)}{dt} = \frac{U-u}{\tau_f} + U(1-u)R = 0 \tag{8}$$

$$\frac{db(t)}{dt} = -\frac{b}{\tau_b} + \lambda_{cb}(B-b)c - \lambda_{bc}b = 0 \tag{9}$$

$$\frac{dc(t)}{dt} = -\frac{c}{\tau_c} - \lambda_{cb}(B-b)c + \lambda_{bc}b + \mu R = 0 \tag{10}$$

Denoting the fixed point by subscript *, the first pair of equations yield

$$u_* = U_* \frac{1+R\tau_f}{1+U_*R\tau_f} \tag{11}$$

$$x_* = \frac{1}{1+u_*R\tau_d} \tag{12}$$

The third equation leads to

$$b_* = B \frac{\tau_b \lambda_{cb} c_*}{1+\tau_b(\lambda_{bc}+\lambda_{cb}c_*)} \tag{13}$$

which is the relation we used to plot Fig 5B. Substituting this relation to the last equation above results in the following parabolic equation for c:

$$c^2[\tau_b\lambda_{cb}] + c[(1+k_c) + k_b(B-\mu R\tau_b)] - \mu R\tau_c(1+k_c) = 0 \tag{14}$$

where we have introduced $k_b = \tau_c\lambda_{cb}$ and $k_c = \tau_b\lambda_{bc}$ for convenience. This parabolic equation has two roots. Yet, only the stable one is of our interest. We numerically check the stability by increasing (or decreasing) the root and asserting that the time derivative of b is negative (or positive), thus exerting an attracting force on c toward the stable root. The stable (and positive) solution, $c_*$, is then

substituted in Equations (11), (12), and (13) for different parameter values to plot Fig 5C.

## Parameter selection

For $\tau_d$ and $\tau_f$ we followed the values reported in Mongillo et al (2008) (200 and 1,500 ms), respectively. The bound and free time scales $\tau_c$ and $\tau_b$ are related to the calcium lifetimes. Because of the high stability of the bound $Ca^{2+}$ pool we choose a value of 10,000 s (corresponding to a time scale of order of days) for $\tau_b$. In contrast, free calcium has a faster turnover. Therefore, we set $\tau_b$ to 100 s. The transition rates $\lambda_{cb}$ and $\lambda_{bc}$ are unknown. However, because of the abundance of bound $Ca^{2+}$ in cells, we assumed that transition to bound states is more likely, and thus promoted binding over unbinding by setting $\lambda_{cb} \gg \lambda_{bc}$. For Fig 5C, values of 200 and 30 are used. Interestingly, as long as the relative ratio of these transition rates is similar, the model is insensitive to the exact values. The stimulation rate R was set to 20 Hz to match the experimental value. Other parameters (B, $\mu$, $\lambda$) were varied either systematically (Fig 5C and D) or manually to align to experimental data.

## Data and Materials Availability

All data are available in the main text or the supplementary materials.

## Supplementary Information

## Acknowledgements

We would like to thank the following current and former members of the Rizzoli group at the University Medical Center Göttingen for their assistance with various aspects of this research: Katharina Grewe and Paola Agüi Gonzalez for assistance with NanoSIMS analysis, Tal Dankovich for assistance with immunostaining, Eugenio Fornasiero for assistance with Nikon imaging, Christina Zeising for assistance with neuronal sample preparation, Sinem Sertel for assistance with the $Ca^{2+}$-imaging protocol, and Selda Glowacki for discussions during the development of the cell culture protocol for $Ca^{2+}$ experiments. This work was funded by the German Research Foundation (Deutsche Forschungsgemeinschaft) under Germany's Excellence Strategy - EXC 2067/1- 390729940 to SO Rizzoli. This work was also funded by German Research Foundation via SFB1286/A03 and 1963/7-3 to SO Rizzoli. Lastly, this work was funded by the German Research Foundation via SFB1286, projects C01 and Z01, as well as project 492788807, to C Tetzlaff.

## Author Contributions

EA Bonnin: conceptualization, resources, data curation, formal analysis, validation, investigation, visualization, methodology, and writing—original draft, review, and editing.
A Golmohammadi: software, formal analysis, and writing—original draft, review, and editing.

R Rehm: investigation, methodology, and writing—review and editing.
C Tetzlaff: conceptualization, software, formal analysis, supervision, funding acquisition, investigation, methodology, and writing—original draft, review, and editing.
SO Rizzoli: conceptualization, resources, data curation, formal analysis, supervision, funding acquisition, validation, investigation, visualization, methodology, project administration, and writing—original draft, review, and editing.

## Conflict of Interest Statement

The authors declare that they have no conflict of interest.

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
