## [Reviewer comments · Life Science Alliance]

Life Science Alliance

High-resolution analysis of bound Ca²⁺ in neurons and synapses

Elisa Bonnin, Arash Golmohammadi, Ronja Rehm, Christian Tetzlaff, and Silvio Rizzoli

DOI: <https://doi.org/10.26508/lsa.202302030>

Corresponding author(s): *Elisa Bonnin, Universitätsmedizin Göttingen and Silvio Rizzoli, University of Göttingen Medical Center*

Review Timeline:	Submission Date:	2023-03-07
	Editorial Decision:	2023-04-06
	Revision Received:	2023-08-11
	Editorial Decision:	2023-09-27
	Revision Received:	2023-10-02
	Accepted:	2023-10-02

Transaction Report:

April 6, 2023

Re: Life Science Alliance manuscript #LSA-2023-02030-T

Dr. Elisa A. Bonnin
University Medical Center Göttingen
Department of Neuro- and Sensory Physiology
Göttingen 37073
Germany

Dear Dr. Bonnin,

Thank you for submitting your manuscript entitled "High-resolution analysis of bound Ca²⁺ in neurons and synapses" to Life Science Alliance. The manuscript was assessed by expert reviewers, whose comments are appended to this letter. We invite you to submit a revised manuscript addressing the Reviewer comments.

Thank you for this interesting contribution to Life Science Alliance. We are looking forward to receiving your revised manuscript.

Sincerely,

B. MANUSCRIPT ORGANIZATION AND FORMATTING:

Reviewer #1 (Comments to the Authors (Required)):

This manuscript combines fluorescence microscopy, NanoSIMS analysis and computational models to assess the role of bound calcium in neurons and synapses. Specifically, fluorescence microscopy was used to visualize immunolabeled components in the synapse and NanoSIMS was used to image the bound calcium at the same location. Correlated analyses revealed a very weak correlation between immunolabeled postsynaptic density protein-95 (PSD95) and bound calcium in the postsynaptic excitatory synapses, and a weak correlation between immunolabeled vesicular GABA transporter (VGAT) and bound calcium in the pre-synapses of inhibitory neurons. A weak anticorrelation between bound calcium and synaptotagmin (Syph) was also found, which the authors say suggests synaptic activity decreased when bound calcium was high. This suggestion is supported by a computational model. I did not understand the development of this computational modeling so will not comment on it. The main strength of the manuscript is the novelty of imaging bound calcium in synapses with a NanoSIMS. The main weakness is the correlations and anti-correlations found are weak and their importance seems to be exaggerated. In addition, the limited explanations of the neurobiology may limit the readership of the article. However, I believe that the authors can address these weaknesses. Specific suggestions are provided below.

1. Several different types of synapses and/or synapse locations are mentioned throughout the manuscript. It would be helpful if the authors explained the functional significance of the pre-synapse, post-synapse, excitatory synapse and inhibitory synapse.
2. Third paragraph in the results section discusses calcium counts above or below 2. Is at 2 counts per pixel or 2 counts per ROI? If it's 2 counts per ROI, what is the size of each ROI?
3. Lines 118-119: "This type of heterogeneous distribution is expected, as..." What heterogeneous distribution are the authors referring to? The previous sentence is about differences in the slope of the best fit line on the graph of calcium signal vs nitrogen or sulfur signal, and heterogeneity is not mentioned.
4. Lines 203 - 205. Please explain what the word strength - means in the phrases "pre-synaptic strength" and "post-synaptic strength". I understand that VGLUT1 and VGAT are located in the pre-synapse so there immunofluorescence locates a pre-synapse and perhaps shows its size and abundance. I don't see how this conveys information about strength (according to the common definition of the word strength).
5. Figure 3 C-E. What is the area on the image that was used to create each point of the graph? It seems like each ROIs must be larger than 1 pixel because the caption says N is 130 for C, 104 for D and 126 for E in these images must contain more pixels than that. Do all the data points correspond to regions of the image where cell material is present? And why are there different numbers of data points on graph C, D and E when they're showing the same region on the sample?
6. Figure 4 C- E. It doesn't seem that each data point on the graph represents one pixel because the figure caption says N = 62, so these are 62 points on each plot but there must be more than 62 pixels that correspond to the cell in each image. What is the size of the area that represents a single data point on the graph? How were these ROIs defined? Does each ROI correspond to an area on the cell or do some of the ROIs correspond to areas where no cell material is present?
7. Figure 3D and E figure caption and the text that draws conclusions from these graphs throughout the manuscript. The R-squared for PSD95 fluorescence intensity vs the calcium counts = 0.0572. Regardless of the P value, that R-squared value means that 5.72% of the variation in calcium counts is linearly related to the variation in PSD95 fluorescence intensity. That's a tiny correlation! It seems the authors are blowing this out of proportion throughout the manuscript been in the abstract. The graph clearly shows that most of the correlation is due to the 5 data points where at the calcium counts are greater than 30. The R-squared value for the VGAT fluorescence intensity vs calcium counts is 0.2051, which means 20.51% of the variation in calcium counts is linearly related to the variation in VGAT fluorescence intensity regardless of the P value. Though that's higher than for Ca counts vs PSD95 fluorescence intensity, it's still a weak correlation, and again, the five data points where the calcium counts are above 30 have high leverage and drive this correlation. Mentions of these correlations should be more tempered throughout the manuscript in the abstract.
8. Figure 4C figure caption and the text that draws conclusions from this graph throughout the manuscript. The R squared value

for calcium counts vs fluorescence intensity from synaptotagmin is 0.0809, which means that only 8.09% of the variation in calcium counts is linearly related to the variation in fluorescence intensity from synaptotagmin, regardless of the P value. Again, that's a very weak (anti-)correlation. The text implies this anti-correlation is stronger than it is. Bound calcium only weakly correlates to post-synaptic "strength".

9. The R squared value for calcium counts vs fluorescence intensity from synaptotagmin (0.0809), which relates to post-synaptic "strength", is less than half of the R squared value for calcium counts vs fluorescence intensity from VGAT (0.2051), which relates to pre-synaptic "strength". So please explain why the abstract in manuscript text states that bound calcium correlates to post-synaptic strength but does not correlate to pre-synaptic activity? Perhaps the perceived discrepancy is related to a misunderstanding of what is meant by synaptic strength as opposed to meaning of synaptic activity.

Other minor issues

Line 25 page 1: "and more than 99% percent being..." The word percent is redundant.

Figure 2A and the caption for figure 2 says the measurements were made at DIV17 but lines 148 and 499 say they were made at DIV18. Which is correct?

Technical question: does a NanoSIMS instrument actually detect Ca^{2+} , which would have $m/z = 402 = 20$, or is it detecting the ion at $m/z = 40$ as a proxy for divalent calcium ion?

Please make the font for the R-squared and P values shown on the graphs in figures 3 and 4 larger. They are very difficult to read.

Reviewer #2 (Comments to the Authors (Required)):

The authors describe a method to measure protein-bound calcium (Ca^{2+}). They propose nanoscale secondary ion mass spectrometry (NanoSIMS) as a new imaging tool to correlate bound Ca^{2+} to biological processes, such as synaptic activity. NanoSIMS analyses can be combined with fluorescent imaging of distinct compartments as well as dynamic assessment of free Ca^{2+} .

Life Science Alliance publishes methodological papers, therefore the submitted article is within the scope of the journal. Having that said, I believe that the authors can make an extra effort to strengthen their study. Correlative analyses may be good, but mechanistic findings are better. Thus, it would be important that the authors test and validate their method by altering the cellular system, hence showing that NanoSIMS can really detect meaningful differences. In this regard, the authors could modulate the expression of Ca^{2+} binding proteins and/or presynaptic components and measure changes in protein-bound calcium vs Ca^{2+} dynamics.

Furthermore, I wonder whether NanoSIMS can be helpful in detecting new Ca^{2+} binding proteins that have not been previously described or are poorly characterized.

Finally, how far can this new method go? Can it be employed only in culture cells? Can the authors use NanoSIMS also in tissues?

As a last recommendation, authors should adjust their references. The field of Ca^{2+} signaling has evolved significantly, therefore it would be important to substitute some of the very old references with new ones, especially when they are reviews.

Reviewer #3 (Comments to the Authors (Required)):

Bonnin et al present a novel and very interesting approach in exploring in neurons and synapses the role of Ca^{2+} in its bound form. This is conceptually a very interesting paper in which authors analyze Ca^{2+} in the bound form in neurons in culture using NanoSIMS and correlate these measurements with diverse aspects of neuronal physiology. Authors explore interesting questions regarding the relationship between Ca^{2+} in its bound form and diverse parameters in neurons: synaptic vesicle recycling, comparative measurement of Ca^{2+} bound in different compartments of neurons and in different synapse types in cultures (i.e. glutamatergic vs gabaergic). Lastly, they proposed a mathematical model that explores the role of bound Ca^{2+} in synapse function.

The approach is highly interesting and, while challenging, it can start giving answers about the role of bound Ca^{2+} in neurons and synapses. However, as it is, experiments presented in the paper may not fully convince readers in the neuroscience field and additional evidence/discussion should be shown.

General comments:

- Given that the culture is a mixture of glia and neurons, are the authors concerned about having a contamination from Ca^{2+} in the

bound form in astrocytes when measuring neurons and synapses that are together with them? A word discussing why this may be a problem is needed.

- The theoretical model is very interesting and provides insights on how one expects synapses to behave depending on the amount of Ca^{2+} in the bound form. However, part of the assumption is based on the fact that bound Ca^{2+} pool is not merely inert, but rather is an important regulator of synaptic activity, implying a direct cross-talk between both free and bound pools. My major concern here is: what is the main source of Ca^{2+} bound signals in synapses? Or in somas? Given the high amount of free Ca^{2+} and phosphate present in the mitochondrial matrix, it has been proposed that precipitation of calcium phosphate in mitochondrial matrix continuously happens, generating high amounts of CaP . Given that mitochondria heavily decorate somas, dendrites, axons and synapses, it is unclear if the signals authors obtained in synapses truly arise from Ca^{2+} -bound in the cytosol. Authors should clarify the possibility of this point when conciliating experimental data and the model.

- Given the novelty and high interest of the research shown here, I would encourage authors to add supplementary data showing more examples of the experiments. This would help the reader understanding the advantages and limitations of the interesting approach presented by the authors.

- Given the importance of technical aspects of the work, I would suggest that methods are extended, particularly those on the analysis of free Ca^{2+} transients using GECIs as well as conditions for culturing neurons.

Below some specific comments on different parts of the manuscript:

Data from Figure 2:

- While I understand the challenge these experiments present, the data presented in figure 2 is heavily undersampled, making it difficult to obtain robust conclusions. Is it possible to repeat the experiment to clarify whether the lack of correlation is not simply a consequence of undersampling? If reproduced, this ideally should be done with an established GECI like GCaMP6 (for which viruses are readily available as well) with a faster imaging rate, which would allow authors to differentiate bursts more robustly. If experiments are too costly to reproduce, authors should mention that undersampling is an issue for conclusions obtained from figure 2.

- Related to figure 2, what does "Total Intensity F/F0" means? Is it a readout of the response of the entire recording?

- Similarly, Peak height appears mislabeled (i.e. if it was F/F0 it should have values like those in Fig. 2D). Related to this, it seems very hard from the data shown in Fig 2D to be able to decide what the baseline is (given that there are continuous fluctuations), which will heavily affect the values of the peaks. Similarly the burst length seems also arbitrary according to what is presented, and I could not find in methods the explanation on how this was decided. More surprisingly, data presented here in Fig 4D does not show the same burst pattern authors have found using this indicator in a similar preparation (Sertel SM 2021a).

- Could the authors extend a bit more on why Ca^{2+} peak per cell were normalized to K^{+} and not to 32S or 14N as in figure 1? I think this is very interesting to neuroscience readers which will be typically not versed in the use of NanoSIMS.

- Data is not shown to support this conclusion in line 191 "This suggests that the free Ca^{2+} fluorescence-derived measurements are not strongly correlated to the bound Ca^{2+} , at the whole-cell level." Baseline free Ca^{2+} signals from GECIs will be heavily affected by the expression levels of the sensor. When using lentivirus one expects relative variability in the expression of the sensor, thus giving the authors variability in F that simply arises from how much GECI there is, and not how much free Ca^{2+} . Therefore a lack of correlation may arise simply because the readout of free Ca^{2+} is not normalized. Normalization of the sensor can be achieved by saturation, which also would provide the authors with a quantitative estimation of how much free Ca^{2+} there is (at least in the cytosol, which is what is being measured). See Maravall Biophys J doi: 10.1016/S0006-3495(00)76809-3 for example. Saturation can be achieved with pharmacology i.e. using high ionomycin concentrations. Alternative, authors could use a ratiometric Ca^{2+} sensor.

- The images from B and C do not seem to be showing the very same neurons. Do the authors have other examples to show with a higher correlation of the somas (i.e. like in Fig. 3)

Data from Figure 3.

- Staining patterns for vGluT and vGat are unusual for cultured neurons (fig 3). The authors don't mention here why this does not look like a typical vGluT/vGAT/PSD95 staining (i.e. en-passant boutons closely apposed to PSD95, for example). Is it because the necessary steps needed for NanoSIMS affect samples in a way that en-passant boutons cannot be seen easily? Given that the expectation would be to see synapses much more clearly, and not necessarily synapses onto a soma (as it looks in Fig 3), authors should make clear what we are supposed to be seeing and why it does not look like the classic staining published in many occasions by many labs (including other publications of the Rizzoli lab). On this regard, it would help to add a supplementary file to show at least 5 more representative examples of these experiments, giving the reader a more clear idea on how raw data looks and the inherent limitations of this approach.

- Fig. 3. It is not clear how many neurons were analyzed to give the individual measurements presented. Authors should clarify this point.

- It is confusing to talk about "synaptic strength" (i.e. line 404) when referring to the "strength" (i.e. intensity) of the staining. In synapse physiology synaptic strength refers to how much information is transmitted per action potential. Thus, these should be rephrased through the text to avoid confusion.

- As inhibitory synapses on average will have a mitochondrion (which has large quantities of Ca^{2+} in the bound form as CaP), compared to excitatory synapses, which typically have a 50% chance to have a mitochondrion, it feels like an interesting point to discuss in the discussion, as this could as well be the reason why that difference is found (and not necessarily the different expression of Ca^{2+} -binding proteins).

Data from Figure 4.

- Given that the much of the 40Ca signal (and the corresponding 14N signal) do not overlap with neuronal staining of Syt1 and NB, could it be that part of the signal comes from astrocytes? I.e. if a region does not have neuronal staining, but 14N and 40Ca staining, can the authors comment on the possible source of this signal?

- It would help to choose an image in which staining for Synaptophysin looks more as typically expected i.e. marking actual presynaptic sites, as depicted in the graphical scheme of A. Also, area shown could be larger to give readers a better idea. Ideally, authors also add a supplementary figure with 5 additional examples to give readers a good sense of how experiments look like. Given that this approach is highly innovative, such effort into showing many examples should help in establishing this technique in the field.

Minor points.

Is data in Figure 2D real or some graphical representation? The authors do not mention the frequency of imaging, which makes it difficult to understand how these experiments were performed.

Could the authors comment on their selection of a non-optimal genetically-encoded Ca²⁺ sensor for the measurements in figure 2? It seems that using GCaMP6 (or newer versions) would give a much better signal-to-noise, allowing to measure with higher quality the parameters of figure 2. In fact, the changes in $\Delta F/F$ in Fig2.D are minuscule, which makes one wonder whether those oscillations could be noise. Typically in culture one has bursts of activity that come back to a stable baseline, to then be able to burst again. It would be worth explaining the need of using a sensor that does not provide signals as typically seen in the field.

Fig 1 is missing scale bar. Ideally authors should add a calibration bar to understand what the chosen LUT means (i.e. is light yellow showing more or less Ca²⁺?)

Line 52 - While it is well established that this is the case, it would help to add a citation that supports this phrase: "The vast majority (>99%) of intracellular Ca²⁺ is in the bound form."

Reply to the Reviewer Comments

All Reviewer comments are shown in *italics* below, with our answers in normal font.

Reviewer #1

This manuscript combines fluorescence microscopy, NanoSIMS analysis and computational models to assess the role of bound calcium in neurons and synapses. Specifically, fluorescence microscopy was used to visualize immunolabeled components in the synapse and NanoSIMS was used to image the bound calcium at the same location. Correlated analyses revealed a very weak correlation between immunolabeled postsynaptic density protein-95 (PSD95) and bound calcium in the postsynaptic excitatory synapses, and a weak correlation between immunolabeled vesicular GABA transporter (VGAT) and bound calcium in the pre-synapses of inhibitory neurons. A weak anticorrelation between bound calcium and synaptotagmin (Syph) was also found, which the authors say suggests synaptic activity decreased when bound calcium was high. This suggestion is supported by a computational model. I did not understand the development of this computational modeling so will not comment on it. The main strength of the manuscript is the novelty of imaging bound calcium in synapses with a NanoSIMS. The main weakness is the correlations and anti-correlations found are weak and their importance seems to be exaggerated. In addition, the limited explanations of the neurobiology may limit the readership of the article. However, I believe that the authors can address these weaknesses. Specific suggestions are provided below.

We thank the Reviewer for the comments.

1. Several different types of synapses and/or synapse locations are mentioned throughout the manuscript. It would be helpful if the authors explained the functional significance of the pre-synapse, post-synapse, excitatory synapse and inhibitory synapse.

We have included a paragraph in the Introduction to address this (Lines 84-94). Here we copy this paragraph:

“We specifically examine bound Ca^{2+} in three locations: i) the excitatory glutamatergic pre-synapse, ii) the inhibitory GABA-ergic pre-synapse, and iii) the excitatory (glutamatergic) post-synapse. Pre-synapses contain neurotransmitter-loaded synaptic vesicles, which release their contents onto synaptic receptors found in the post-synapse. When neurons are active, the vesicles from the excitatory synapses release glutamate, an amino acid that stimulates several types of ionotropic (ion channels) and metabotropic (non-channel) receptors in the respective post-synapse, thereby causing a depolarization and activation of the cell. The inhibitory vesicles contain GABA, a modified amino acid, which stimulates inhibitory receptors in its post-synapse, leading to a hyperpolarization (inactivation) of the cell (Martin, 2021). Both excitatory and inhibitory neurons are required in order to balance the flow of electrical information in the brain (Kajiwara *et al*, 2021).”

2. Third paragraph in the results section discusses calcium counts above or below 2. Is at 2 counts per pixel or 2 counts per ROI? If it's 2 counts per ROI, what is the size of each ROI?

This has been clarified in the respective paragraph. The meaning is 2 counts per ROI. The size of the ROI has been added to the Methods section; see new lines 767-769, also quoted below:

“ROIs are defined as circles with a radius of 6 pixels, leading to an interior area of ~113 pixels. Intensity values for each pixel within an ROI are averaged, so that a single average value per ROI is recorded.”

3. Lines 118-119: *"This type of heterogeneous distribution is expected, as..."* What heterogeneous distribution are the authors referring to? The previous sentence is about differences in the slope of the best fit line on the graph of calcium signal vs nitrogen or sulfur signal, and heterogeneity is not mentioned.

We are referring to the following: two groups of signals can be observed in the respective graph, one with high calcium levels, and one with low calcium levels. The calcium signals in both of these groups correlate to the levels of cellular proteins, represented by nitrogen and sulfur. These groups were described in the sentences before the one mentioned by the Reviewer.

We have rephrased this statement, to be clearer, and it now reads:

“The existence of these two distinct groups of signals is expected, as Ca²⁺-binding proteins are not thought to be present at equal levels in all cellular regions. “

4. Lines 203 - 205. Please explain what the word strength means in the phrases "pre-synaptic strength" and "post-synaptic strength". I understand that VGLUT1 and VGAT are located in the pre-synapse so there immunofluorescence locates a pre-synapse and perhaps shows its size and abundance. I don't see how this conveys information about strength (according to the common definition of the word strength).

Synaptic strength is defined as the amplitude of the signal produced by the respective synapse in a network. As this type of measurement is not available for immunostaining experiment, the strength of the synapse is estimated by the size of the pre-synaptic vesicle pool, or by the size of the post-synaptic density (as discussed, for example, by Humeau & Choquet, 2019). Following this type of argument, the intensity of the VGLUT1 or VGAT signals is a direct estimation of synaptic strength.

However, we agree with the Reviewer that this type of terminology is not necessary for our manuscript. We therefore changed all references to “synaptic strength” to the following: “the intensity of the VGLUT1 signal”, “VGAT signal” or “PSD95 signal”.

5. Figure 3 C-E. What is the area on the image that was used to create each point of the graph? It seems like each ROIs must be larger than 1 pixel because the caption says N is 130 for C, 104 for D and 126 for E in these images must contain more pixels than that.

The size of the ROI has been added to the Methods section (see new lines 767-769, also quoted below):

“ROIs are defined as circles with a radius of 6 pixels, leading to an interior area of ~113 pixels. Intensity values for each pixel within an ROI are averaged, so that a single average value per ROI is recorded.”

Do all the data points correspond to regions of the image where cell material is present?

Yes, this is true. In that same section (lines 756-760), we initially wrote the following:

“For the VGAT/PSD95/VGLUT1 image set (Fig. 3), ROIs were selected based on high signal either in VGAT or in VGLUT, denoting either an inhibitory or excitatory synapse respectively.”

To this line we now added, for clarity: “or based on high signal in PSD95, denoting the location of the excitatory post-synapse”.

And why are there different numbers of data points on graph C, D and E when they're showing the same region on the sample?

As mentioned in our previous lines, we did not use the same region on the sample for each of graphs C, D, and E. For Figure 3, graph C, which examines VGLUT, we selected only points that contained VGLUT (thus, were VGLUT-positive), denoting the location of an excitatory synapse. This is because VGLUT marks the location of an excitatory synapse, and we wanted to examine the behavior of 40Ca at these locations.

Similarly, for Figure 3, graph E, we were only looking at areas that were VGAT-positive, because we were specifically examining inhibitory synapses. Finally, for graph D, we examined areas that were PSD95-positive, because we wanted to examine the postsynapse. The reference to the same region on the sample refers to the fact that each point in Graph C is the same ROI in both the VGAT and 40Ca image, the same point in Graph D is the same ROI in both the PSD95 and 40Ca image, and the same point in Graph E is the same ROI in both the VGAT and 40Ca image.

6. Figure 4 C- E. It doesn't seem that each data point on the graph represents one pixel because the figure caption says $N = 62$, so these are 62 points on each plot but there must be more than 62 pixels that correspond to the cell in each image. What is the size of the area that represents a single data point on the graph? How were these ROIs defined? Does each ROI correspond to an area on the cell or do some of the ROIs correspond to areas where no cell material is present?

The size of the ROI has been added to the methods section (see new lines 767-769, also quoted below):

“ROIs are defined as circles with a radius of 6 pixels, leading to an interior area of ~113 pixels. Intensity values for each pixel within an ROI are averaged, so that a single average value per ROI is recorded.”

In that same section (lines 754-772) we clarify that each ROI was hand-selected, so that only locations that correspond to cellular material were selected. The new text is shown in bold, below:

“ROIs were selected **by hand** based on high signals in fluorescence. For comparison between ^{14}N , ^{32}S , and ^{40}Ca (Fig. 1), ROIs were selected based on high signal in either ^{14}N or ^{32}S . For the VGAT/PSD95/VGLUT1 image set (Fig. 3), ROIs were selected based on high signal either in VGAT or in VGLUT, denoting either an inhibitory or excitatory synapse respectively, or based on high signal in PSD95, denoting the location of the excitatory post-synapse. For the Syt1/NB/Syph image set (Fig. 4), ROIs were selected based on high intensity in any of the three fluorescent markers. **Locations that did not correspond to cellular material were excluded from analysis.**

The ROI selection script works by asking the user to select background points at each image. Once background levels are recorded, the user can select ROIs. The script then records the corresponding intensity value at each selected location for each associated image, subtracted from background. ROIs are defined as circles with a radius of 6 pixels, leading to an interior area of ~113 pixels. Intensity values for each pixel within an ROI are averaged, so that a single average value per ROI is recorded. Prior to further analysis data points with ^{40}Ca intensities at or below background level were removed. This was done by assuming a normal distribution of background ^{40}Ca intensities with a mean of 0 and filtering out points within one standard deviation of the mean.”

We now also include a supplementary figure (Fig. S8) to show how the ROI selection script operates. We are also including the Matlab scripts, as a supplementary Zip file (Data S4).

7. Figure 3D and E figure caption and the text that draws conclusions from these graphs throughout the manuscript. The R-squared for PSD95 fluorescence intensity vs the calcium counts = 0.0572. Regardless of the P value, that R-squared value means that 5.72% of the variation in calcium counts is linearly related to the variation in PSD95 fluorescence intensity. That's a tiny correlation! It seems the authors are blowing this out of proportion throughout the manuscript been in the abstract. The graph clearly shows that most of the correlation is due to the 5 data points where at the calcium counts are greater than 30. The R-squared value for the VGAT fluorescence intensity vs calcium counts is 0.2051, which means 20.51% of the variation in calcium counts is linearly related to the variation in VGAT fluorescence intensity regardless of the P value. Though that's higher than for Ca counts vs PSD95 fluorescence intensity, it's still a weak correlation, and again, the five data points where the calcium counts are above 30 have high leverage and drive this correlation. Mentions of these correlations should be more tempered throughout the manuscript in the abstract.

We agree with the Reviewer. We made the necessary changes throughout the text.

8. Figure 4C figure caption and the text that draws conclusions from this graph throughout the manuscript. The R squared value for calcium counts vs fluorescence intensity from

synaototagmin is 0.0809, which means that only 8.09% of the variation in calcium counts is linearly related to the variation in fluorescence intensity from synaototagmin, regardless of the P value. Again, that's a very weak (anti-)correlation. The text implies this anti-correlation is stronger than it is. Bound calcium only weakly correlates to post-synaptic "strength".

We have now repeated our analysis of synaptotagmin correlations, increasing the number of data points by more than 10-fold. The correlations persisted, and their significance became much stronger (extremely low *p* values). The R squared value remains small, as the Reviewer noted. However, the fact that a highly significant correlation can be measured is important.

We adjusted our text, as suggested by the Reviewer, but we feel, nonetheless, that it is important to communicate this newly-discovered correlation to the neuronal cell biology field.

We have now added this explanation to our text, lines 358-362:

"While the correlations between the Syt1 and NB labeling and bound Ca²⁺ have low R² values (Fig. 4C,D), they are highly significant statistically, and are visible even on small subsets of the analyzed data. Thus, while variation in bound Ca²⁺ does not account for a substantial proportion of the dynamics of synaptic activity, it is nonetheless a factor that probably influences the synapse, as also explored in the following section."

9. The R squared value for calcium counts vs fluorescence intensity from synaototagmin (0.0809), which relates to post-synaptic "strength", is less than half of the R squared value for calcium counts vs fluorescence intensity from VGAT (0.2051), which relates to pre-synaptic "strength". So please explain why the abstract in manuscript text states that bound calcium correlates to post-synaptic strength but does not correlate to pre-synaptic activity? Perhaps the perceived discrepancy is related to a misunderstanding of what is meant by synaptic strength as opposed to meaning of synaptic activity.

We apologize for this confusion. Synaptotagmin does not relate to post-synaptic strength. This is a pre-synaptic vesicle molecule, which we label with antibodies during synaptic activity. The labeling is performed under saturating conditions, so that all active pre-synaptic vesicles are labeled. Therefore, this labeling procedure measures the amounts of active vesicles present in each synapse. The correlation of this label to Ca²⁺ is negative, as shown in Fig. 4. The correlation of Ca²⁺ to post-synaptic strength (determined by the PSD95 signal) is positive, as shown in Fig. 3. This explains the respective phrasing in our abstract.

Other minor issues:

Line 25 page 1: "and more than 99% percent being..." The word percent is redundant.

This line has been revised based on the reviewer's comments.

Figure 2A and the caption for figure 2 says the measurements were made at DIV17 but lines 148 and 499 say they were made at DIV18. Which is correct?

DIV17 is correct. Line 163 has been revised accordingly. Line 596 refers to a second set of neurons different from the one mentioned in that analysis, which were indeed analyzed on DIV18.

Technical question: does a NanoSIMS instrument actually detect Ca^{2+} , which would have $m/z = 402 = 20$, or is it detecting the ion at $m/z = 40$ as a proxy for divalent calcium ion?

It is technically detecting the ion at $m/z = 40$, which is why this ion is referred to as ^{40}Ca in all graphs where data is shown. We are using this ion at $m/z = 40$ as a proxy for the divalent calcium ion.

Please make the font for the R-squared and P values shown on the graphs in figures 3 and 4 larger. They are very difficult to read.

These images have been edited.

Reviewer #2

The authors describe a method to measure protein-bound calcium (Ca^{2+}). They propose nanoscale secondary ion mass spectrometry (NanoSIMS) as a new imaging tool to correlate bound Ca^{2+} to biological processes, such as synaptic activity. NanoSIMS analyses can be combined with fluorescent imaging of distinct compartments as well as dynamic assessment of free Ca^{2+} . Life Science Alliance publishes methodological papers, therefore the submitted article is within the scope of the journal. Having that said, I believe that the authors can make an extra effort to strengthen their study. Correlative analyses may be good, but mechanistic findings are better.

We thank the Reviewer for the comments.

Thus, it would be important that the authors test and validate their method by altering the cellular system, hence showing that NanoSIMS can really detect meaningful differences. In this regard, the authors could modulate the expression of Ca^{2+} binding proteins and/or presynaptic components and measure changes in protein-bound calcium vs Ca^{2+} dynamics.

As the Reviewer is probably aware, the field of Ca^{2+} buffering in neurons is a very challenging one. First, we are unsure about the identities of the precise Ca^{2+} -binding proteins that make the strongest differences in our system. Therefore, it is difficult to know which molecules should be manipulated. Also, the manipulation of such molecules by knock-down procedures would probably injure the neurons, since Ca^{2+} is an essential second messenger for these cells, and interfering with Ca^{2+} buffering would modify a plethora of cellular functions. In the experience of the last author, increasing free Ca^{2+} concentrations in the neuronal somas of cultured neurons, even transiently, to around $5 \mu\text{M}$ resulted in cell death within a few hours (work related to Rizzoli *et al*, 2002).

Second, the Ca²⁺-buffering proteins that are often discussed in the literature, as parvalbumin, are extremely abundant, so that over-expression, using common plasmid- or virus-derived transfection methods would lead only to mild increases, therefore not providing meaningful changes that we could study.

Nonetheless, the Reviewer is right in pointing out that more experimental insight would be valuable. We decided, therefore, to tackle the issue of Ca²⁺ turnover in cells, meaning the rate of exchange of these ions. In principle, free Ca²⁺ enters and leaves the cells through channels and transporters permanently, implying that a rapid turnover of this pool is expected. However, little information has been obtained relating to bound Ca²⁺. In principle, a small percentage of the ionic composition of the neuron will exchange after each action potential. With action potential bursts happening every few seconds in culture, strong changes would be expected after hours or days of activity.

To test this, we pulsed cells with isotopic calcium, ⁴⁶Ca²⁺, and measured the appearance of the isotopes in the bound Ca²⁺ pool. This process was very slow for fibroblasts (HEK cells), only reaching, after 3 days of pulsing, approximately 4% of the maximum possible isotopic labeling. This is remarkable, since the fibroblasts also divided during this time interval, and therefore needed to produce new calcium-binding proteins.

Ca²⁺ turnover was also slow in neurons, reaching approximately 20% of the maximum labeling after 10 days of pulsing. This implies that, in spite of rapid dynamics for free Ca²⁺, the bound population is remarkably stable, both in neurons and in rapidly-growing cells as fibroblasts. This is all the more surprising, since the lifetimes of Ca²⁺ buffer proteins as calbindin, calretinin and parvalbumin are not particularly long (close to the median of all neuronal proteins, Fornasiero *et al*, 2018). Overall, this finding implies that the exchange of this ion between the cell and its external milieu is, overall, very slow, a novel finding for the neuroscience field.

The section dealing with Ca²⁺ turnover has been added to the main text in lines 216-235, lines 703-742, and in the supplementary figures S2 and S3.

Furthermore, I wonder whether NanoSIMS can be helpful in detecting new Ca²⁺ binding proteins that have not been previously described or are poorly characterized.

Unfortunately, one of the limitations of NanoSIMS is that it does not provide information on the molecular level. Because the NanoSIMS technique works by breaking the bonds between atoms, it cannot specifically tell us which protein is binding Ca²⁺.

Finally, how far can this new method go? Can it be employed only in culture cells? Can the authors use NanoSIMS also in tissues?

NanoSIMS can be readily used in tissues, as we showed in the past for different questions (Dankovich *et al*, 2021; Bonnin *et al*, 2021; Li *et al*, 2022). There is no limitation in using the calcium-imaging approach in tissues.

Nonetheless, we did explore potential limitations, to answer the Reviewer thoroughly. We have recently performed tissue imaging by correlative NanoSIMS and transmission electron

microscopy (TEM), as published in Michanski *et al*, 2023). To test the limits of our approach, we explored the use of TEM-NanoSIMS for Ca²⁺ imaging. The TEM samples are extremely fragile, and as the NanoSIMS beam destroys each sample, in order to measure it. This implies that a very careful NanoSIMS tuning is required to measure negative ions (like those derived from carbon, nitrogen, sulfur, etc.) in TEM samples, using the “positive mode” Cs⁺ beam of the instrument. The “negative mode” O⁻ beam of the NanoSIMS instrument, used to measure positive ions like Ca²⁺, has an approximately 6-fold higher intensity than the Cs⁺ beam. This resulted in the destruction of the TEM samples, before they could provide meaningful information. This is an unexpected limitation, which, however, could be avoided by replacing TEM with scanning electron microscopy (SEM), in which we could employ the exact same samples as used in the rest of our work (200 nm sections, on silicon support).

As a last recommendation, authors should adjust their references. The field of Ca²⁺ signaling has evolved significantly, therefore it would be important to substitute some of the very old references with new ones, especially when they are reviews.

We have updated the reference list. Some of the older references have been retained, especially those relevant for the discussion of methods. Most of the other old references have been replaced with newer ones.

Reviewer #3

Bonnin et al present a novel and very interesting approach in exploring in neurons and synapses the role of Ca²⁺ in its bound form. This is conceptually a very interesting paper in which authors analyze Ca²⁺ in the bound form in neurons in culture using NanoSIMS and correlate these measurements with diverse aspects of neuronal physiology. Authors explore interesting questions regarding the relationship between Ca²⁺ in its bound form and diverse parameters in neurons: synaptic vesicle recycling, comparative measurement of Ca²⁺ bound in different compartments of neurons and in different synapse types in cultures (i.e. glutamatergic vs gabaergic). Lastly, they proposed a mathematical model that explores the role of bound Ca²⁺ in synapse function. The approach is highly interesting and, while challenging, it can start giving answers about the role of bound Ca²⁺ in neurons and synapses. However, as it is, experiments presented in the paper may not fully convince readers in the neuroscience field and additional evidence/discussion should be shown.

We thank the Reviewer for the comments.

General comments:

- Given that the culture is a mixture of glia and neurons, are the authors concern about having a contamination from Ca²⁺ in the bound form in astrocytes when measuring neurons and synapses that are together with them? A word discussing why this may be a problem is needed.

The axial resolution of the NanoSIMS is around 10-20 nm in our experiments (Saka *et al*, 2014). Therefore, we do not provide “mixed” measurements of multiple overlapping cells. In other words, when measuring from a region marked by neuronal or synaptic proteins, we are certain to provide ionic values from a neuronal cell. We have added this explanation to the manuscript on lines 252-256.

- The theoretical model is very interesting and provides insights on how one expects synapses to behave depending on the amount of Ca^{2+} in the bound form. However, part of the assumption is based on the fact that bound Ca^{2+} pool is not merely inert, but rather is an important regulator of synaptic activity, implying a direct cross-talk between both free and bound pools. My major concern here is: what is the main source of Ca^{2+} bound signals in synapses? Or in somas? Given the high amount of free Ca^{2+} and phosphate present in the mitochondrial matrix, it has been proposed that precipitation of calcium phosphate in mitochondrial matrix continuously happens, generating high amounts of CaP . Given that mitochondria heavily decorate somas, dendrites, axons and synapses, it is unclear if the signals authors obtained in synapses truly arise from Ca^{2+} -bound in the cytosol. Authors should clarify the possibility of this point when conciliating experimental data and the model.

The data we have is not compatible with phosphate-bound Ca^{2+} ions. The levels of phosphorus (representing phosphate) are not measurable in our samples, presumably due to the plastic embedding procedure we follow, which will remove free salts. This is shown in the image below, where phosphorus is only visible at very low levels outside the nucleus, where it is present in DNA molecules. Please see below, in Fig. 1 (for Reviewers only).

Figure 1 (for Reviewers only). Phosphorus and nitrogen imaging with NanoSIMS. (A) ^{31}P image containing three neuron nuclei from an embedded sample. Brighter colors indicate higher ^{31}P intensity. The ^{31}P signal is generally low, but where present, is concentrated around the nucleus. Scale bar = 10 μm . **(B)** ^{14}N intensity image of the same location taken simultaneously with the ^{31}P image. Cellular features are more visible in ^{14}N . Brighter colors indicate higher ^{14}N intensity. Scale bar = 10 μm .

At the same time, the Ca^{2+} signals are not compatible with mitochondria, taking into account their appearance in NanoSIMS imaging. Mitochondria appear as hard, round objects, or networks of objects, in NanoSIMS, which is completely different from the diffuse appearance of Ca^{2+} signals in our present work. Please see below, in Fig. 2 (for Reviewers only), some images of mitochondria labeled in NanoSIMS. Please refer also to our previous work (Agüi-Gonzalez *et al*, 2021) for images of mitochondria labeled in neurons, which show similar patterns.

Overall, this implies that the source of Ca^{2+} is probably in the cellular cytosol, and possibly also in the endoplasmic reticulum, in the form of proteins binding this ion.

Figure 2 (for Reviewers only). Mitochondria imaging in NanoSIMS. (A) To recognize individual organelles in nanoSIMS, they need to be immunostained with probes containing special isotopes. We immunostained here mitochondria in fibroblasts, using ^{19}F -coupled probes. The “hot” colors indicate mitochondria. Scale bar = 5 μm . Published in Kabatas *et al*, (2019). **(B)** Click chemistry can be also used for this approach. Here mitochondria are labeled in only one cell, in which the mitochondria marker TOM70 was expressed with an unnatural amino acid (UAA) inserted in its sequence. The ^{19}F -coupled probe was then coupled to the UAA by a copper-catalyzed reaction. One single cell was transfected, and the mitochondria are strongly labeled (“hot colors”). Scale bar = 10 μm .

- Given the novelty and high interest of the research shown here, I would encourage authors to add supplementary data showing more examples of the experiments. This would help the reader understanding the advantages and limitations of the interesting approach presented by the authors.

We have performed this, please see supplementary figures S4 and S6.

- Given the importance of technical aspects of the work, I would suggest that methods are extended, particularly those on the analysis of free Ca^{2+} transients using GECIs as well as conditions for culturing neurons.

We have extended the Methods section, especially the points noted by the Reviewer.

*Below some specific comments on different parts of the manuscript:
Data from Figure 2:*

- While I understand the challenge these experiments present, the data presented in figure 2 is heavily undersampled, making it difficult to obtain robust conclusions. Is it possible to repeat the

experiment to clarify whether the lack of correlation is not simply a consequence of undersampling?

We repeated these experiments, more than doubling the number of cells analyzed. At the same time, we also included an experimental condition where we stimulate the cells, with both short- and long-term action potential trains, to extract more information. None of our conclusions changed.

If reproduced, this ideally should be done with an established GECI like GCaMP6 (for which viruses are readily available as well) with a faster imaging rate, which would allow authors to differentiate bursts more robustly. If experiments are too costly to reproduce, authors should mention that undersampling is an issue for conclusions obtained from figure 2.

We attempted to work with GCaMP6, which, unfortunately, failed at the step of viral transfection, for multiple experiments (although we did use this construct in the past, Fornasiero *et al*, 2018). We, therefore, repeated the Neuroburst experiments with better imaging quality. The main problem with the experiments presented in the previous version of the manuscript was the use of plastic-bottom dishes for growing the cells, which impeded imaging. We have now grown the cells on glass coverslips, which improved image quality substantially. We are now able to describe bursts very well, as shown in the new Figure 2.

- Related to figure 2, what does "Total Intensity F/F0" means? Is it a readout of the response of the entire recording?

This value means the "area under the peak" for each burst, which is then averaged for a specific cell. Please refer to the graphical legend in Fig. 2K for the definition of the different measurements.

- Similarly, Peak height appears mislabeled (i.e. if it was F/F0 it should have values like those in Fig. 2D). Related to this, it seems very hard from the data shown in Fig 2D to be able to decide what the baseline is (given that there are continuous fluctuations), which will heavily affect the values of the peaks. Similarly the burst length seems also arbitrary according to what is presented, and I could not find in methods the explanation on how this was decided. More surprisingly, data presented here in Fig 4D does not show the same burst pattern authors have found using this indicator in a similar preparation (Sertel SM 2021a).

As mentioned above, we have completely re-made this dataset, using much higher imaging quality. This renders the burst measurements much more straightforward, and more similar to previous work in the field, including our older data (Sertel *et al*, 2021). Please see new Fig. 2.

- Could the authors extend a bit more on why Ca²⁺ peak per cell were normalized to K⁺ and not to 32S or 14N as in figure 1? I think this is very interesting to neuroscience readers which will be typically not versed in the use of NanoSIMS.

NanoSIMS data are often normalized, in many fields of science, since variations in the height of the sample surface can lead to different angles of incidence between the beam and the sample.

This causes different ion outputs, implying that, for such samples, the signals for all ions may vary widely, based on the 3D structure of the specimen. Generating ratios between such signals is then necessary. For Ca^{2+} , which is measured simultaneously with K^+ and Na^+ , a ratio to the more abundant K^+ would make more sense than a ratio to sulfur or nitrogen, which are measured in a separate instrument run, using a different primary ion beam (Cs^+ for sulfur and nitrogen, and O^- for Ca^{2+} , K^+ and Na^+).

Nonetheless, the Reviewer is right in mentioning this issue. In principle, normalization is not even needed for our plastic-embedded samples, since sectioning with a diamond knife leads to a perfectly flat sample. We always collect back-scattered electron images to verify the geometry of the sample (as it is customary in NanoSIMS), so that we can, indeed, state that every sample was flat. Therefore, we removed this unnecessary normalization. This did not change, at all, our conclusions.

Another small point: in Fig. 1, there was no normalization to ^{32}S or ^{14}N . Signals were simply plotted for Ca^{2+} versus ^{32}S and ^{14}N , in the respective figure.

- Data is not shown to support this conclusion in line 191 "This suggests that the free Ca^{2+} fluorescence-derived measurements are not strongly correlated to the bound Ca^{2+} , at the whole-cell level." Baseline free Ca^{2+} signals from GECIs will be heavily affected by the expression levels of the sensor. When using lentivirus one expects relative variability in the expression of the sensor, thus giving the authors variability in F that simply arises from how much GECI there is, and not how much free Ca^{2+} . Therefore a lack of correlation may arise simply because the readout of free Ca^{2+} is not normalized. Normalization of the sensor can be achieved by saturation, which also would provide the authors with a quantitative estimation of how much free Ca^{2+} there is (at least in the cytosol, which is what is being measured). See Maravall Biophys J doi: 10.1016/S0006-3495(00)76809-3 for example. Saturation can be achieved with pharmacology i.e. using high ionomycin concentrations. Alternative, authors could use a ratiometric Ca^{2+} sensor.

We have performed the experiment suggested by the Reviewer, using ionomycin and normalizing the resulting signals accordingly. This left our conclusions unaffected. Please see the new Fig. 2.

- The images from B and C do not seem to be showing the very same neurons. Do the authors have other examples to show with a higher correlation of the somas (i.e. like in Fig. 3)

The sample preparation procedures result in a very different appearance for the cells, which was also affected by the poor image quality of our previous experiments, due to imaging through plastic-bottom dishes.

However, even for the new images, the correlation is not as good as when imaging neurites. NanoSIMS only presents images through a thin layer of the sample, of tens of nanometers, while live imaging of somas was performed with a 20x air objective, imaging the whole micrometers-thick cell. For neurites, they were imaged in fluorescence using a 60x oil-immersion objective, after embedding and sectioning to 200 nm slices, so that the resulting correlation will always be much better. In simple terms, the live-cell/NanoSIMS image cannot be

as well correlated as the fluorescence/NanoSIMS images of the same fixed and plastic-embedded sections.

To account for this, for the live cells we only estimate cell-wide signals in both techniques, while the immunostained samples allow us to measure single objects in both fluorescence and NanoSIMS.

Data from Figure 3.

- Staining patterns for vGlut and vGat are unusual for cultured neurons (fig 3). The authors don't mention here why this does not look like a typical vGlut/vGAT/PSD95 staining (i.e. en-passant boutons closely apposed to PSD95, for example). Is it because the necessary steps needed for NanoSIMS affect samples in a way that en-passant boutons cannot be seen easily?

The immunostaining images generated for these figures are of samples that have been already prepared for NanoSIMS analysis. That means these samples have been embedded in plastic resin and sliced into 200 nm slices. Therefore, we are not viewing these samples in the same way that fixed neurons would be viewed in a confocal microscope, since the axial resolution is a few-fold higher, due to the thickness of the sample. This is an inherent limitation of the NanoSIMS approach, as thicker specimens are very difficult to use.

Given that the expectation would be to see synapses much more clearly, and not necessarily synapses onto a soma (as it looks in Fig 3), authors should make clear what we are supposed to be seeing and why it does not look like the classic staining published in many occasions by many labs (including other publications of the Rizzoli lab). On this regard, it would help to add a supplementary file to show at least 5 more representative examples of these experiments, giving the reader a more clear idea on how raw data looks and the inherent limitations of this approach.

Please see our reply to the previous comment, referring to the appearance of samples processed for NanoSIMS. Please also refer to supplementary figures XX to YY for more images, in which we also made efforts to increase imaging quality.

Imaging quality also depends on the amount of oil present on the samples during fluorescence imaging. This needs to be maintained at minimal levels, to avoid a situation in which the oil-contaminated sample cannot be measured in NanoSIMS. Therefore, only thin films of oil are applied. Therefore, air bubbles between the sample and the objective form occasionally, implying lowered image quality. We now tried to eliminate such images, and add new ones, from carefully performed experiments (e.g. the new Fig. 4).

- Fig. 3. It is not clear how many neurons were analyzed to give the individual measurements presented. Authors should clarify this point.

We have clarified these points. Please see lines 256-259 for the VGAT/VGLUT/PSD95 experiment, and lines 311-313 for the Syt-1/Syph/NB experiment.

- It is confusing to talk about "synaptic strength" (i.e. line 404) when referring to the "strength" (i.e. intensity) of the staining. In synapse physiology synaptic strength refers to how much information is transmitted per action potential. Thus, these should be rephrased through the text to avoid confusion.

We have now removed the term "synaptic strength", and we only refer to the intensities of the respective immunostainings.

- As inhibitory synapses on average will have a mitochondrion (which has large quantities of Ca^{2+} in the bound form as CaP), compared to excitatory synapses, which typically have a 50% chance to have a mitochondrion, it feels like an interesting point to discuss in the discussion, as this could as well be the reason why that difference is found (and not necessarily the different expression of Ca^{2+} -binding proteins).

We have now made a mention of this point in our Discussion, lines 492-496, and quoted below, albeit our data, as explained on pages 9 and 10, above, do not fit with the concept that bound Ca^{2+} is primarily present in mitochondria.

"Importantly, some of the differences noted here between different types of synapses may be due to the presence of mitochondria at higher levels, and with higher frequencies, in particular synaptic compartments (Rossi & Pekkurnaz, 2019). The mitochondria presence would influence strongly local Ca^{2+} buffering, leading to different organizations in the respective Ca^{2+} -binding machineries."

Data from Figure 4.

- Given that the much of the ^{40}Ca signal (and the corresponding ^{14}N signal) do not overlap with neuronal staining of Syt1 and NB, could it be that part of the signal comes from astrocytes? I.e. if a region does not have neuronal staining, but ^{14}N and ^{40}Ca staining, can the authors comment on the possible source of this signal?

The Reviewer is right. Such areas can, indeed, be parts of glia or other neuronal compartments. ^{14}N and ^{40}Ca are ubiquitous in cellular material, so that we need to rely on specific immunostaining to recognize the synaptic areas.

- It would help to choose an image in which staining for Synaptophysin looks more as typically expected i.e. marking actual presynaptic sites, as depicted in the graphical scheme of A. Also, area shown could be larger to give readers a better idea. Ideally, authors also add a supplementary figure with 5 additional examples to give readers a good sense of how experiments look like. Given that this approach is highly innovative, such effort into showing many examples should help in establishing this technique in the field.

Please see the new supplementary figure S6, as well as the new Fig. 4.

- Minor points.

Is data in Figure 2D real or some graphical representation? The authors do not mention the frequency of imaging, which makes it difficult to understand how these experiments were performed.

We now present this in Fig. 2K, which depicts real data. The imaging frequency (1 Hz) is now included in the Methods section.

Could the authors comment on their selection of a non-optimal genetically-encoded Ca²⁺ sensor for the measurements in figure 2? It seems that using GCaMP6 (or newer versions) would give a much better signal-to-noise, allowing to measure with higher quality the parameters of figure 2. In fact, the changes in $\Delta F/F$ in Fig2.D are minuscule, which makes one wonder whether those oscillations could be noise. Typically in culture one has bursts of activity that come back to a stable baseline, to then be able to burst again. It would be worth explaining the need of using a sensor that does not provide signals as typically seen in the field.

Please see our reply to the comments on the Reviewer's section "Data from Figure 2", on page 11. In brief, our imaging quality was not the best, resulting in poor measurements. We have now improved this substantially.

Fig 1 is missing scale bar. Ideally authors should add a calibration bar to understand what the chosen LUT means (i.e. is light yellow showing more or less Ca²⁺?)

This figure has been adjusted.

Line 52 - While it is well established that this is the case, it would help to add a citation that supports this phrase: "The vast majority (>99%) of intracellular Ca²⁺ is in the bound form."

A citation has been added.

References

- Agüi-Gonzalez P, Dankovich TM, Rizzoli SO & Phan NTN (2021) Gold-Conjugated Nanobodies for Targeted Imaging Using High-Resolution Secondary Ion Mass Spectrometry. *Nanomaterials* 11: 1797
- Bonnin EA, Fornasiero EF, Lange F, Turck CW & Rizzoli SO (2021) NanoSIMS observations of mouse retinal cells reveal strict metabolic controls on nitrogen turnover. *BMC Mol and Cell Biol* 22: 5
- Dankovich TM, Kaushik R, Olsthoorn LHM, Petersen GC, Giro PE, Kluever V, Agüi-Gonzalez P, Grewe K, Bao G, Beuermann S, *et al* (2021) Extracellular matrix remodeling through endocytosis and resurfacing of Tenascin-R. *Nat Commun* 12: 7129
- Fornasiero EF, Mandad S, Wildhagen H, Alevra M, Rammner B, Keihani S, Opazo F, Urban I, Ischebeck T, Sakib MS, *et al* (2018) Precisely measured protein lifetimes in the mouse brain reveal differences across tissues and subcellular fractions. *Nat Commun* 9: 4230
- Humeau Y & Choquet D (2019) The next generation of approaches to investigate the link between synaptic plasticity and learning. *Nat Neurosci* 22: 1536–1543
- Kabatas S, Agüi-Gonzalez P, Hinrichs R, Jähne S, Opazo F, Diederichsen U, Rizzoli SO & Phan NTN (2019) Fluorinated nanobodies for targeted molecular imaging of biological samples using nanoscale secondary ion mass spectrometry. *J Anal At Spectrom* 34: 1083–1087
- Kajiwara M, Nomura R, Goetze F, Kawabata M, Isomura Y, Akutsu T & Shimono M (2021) Inhibitory neurons exhibit high controlling ability in the cortical microconnectome. *PLoS Comput Biol* 17: e1008846
- Li F, Fornasiero EF, Dankovich TM, Kluever V & Rizzoli SO (2022) A Reliable Approach for Revealing Molecular Targets in Secondary Ion Mass Spectrometry. *IJMS* 23: 4615
- Martin AR (2021) From neuron to brain Sixth edition. Sunderland, Massachusetts: Oxford University Press/Sinauer
- Michanski S, Henneck T, Mukhopadhyay M, Steyer AM, Gonzalez PA, Grewe K, Ilgen P, Gültas M, Fornasiero EF, Jakobs S, *et al* (2023) Age-dependent structural reorganization of utricular ribbon synapses. *Front Cell Dev Biol* 11
- Rizzoli S, Sharma G & Vijayaraghavan S (2002) Calcium rise in cultured neurons from medial septum elicits calcium waves in surrounding glial cells. *Brain Research* 957: 287–297
- Saka SK, Vogts A, Kröhnert K, Hillion F, Rizzoli SO & Wessels JT (2014) Correlated optical and isotopic nanoscopy. *Nat Commun* 5: 3664
- Sertel SM, von Elling-Tammen MS & Rizzoli SO (2021) The mRNA-Binding Protein RBM3 Regulates Activity Patterns and Local Synaptic Translation in Cultured Hippocampal Neurons. *J Neurosci* 41: 1157–1173

September 27, 2023

RE: Life Science Alliance Manuscript #LSA-2023-02030-TR

Dr. Elisa A. Bonnin
Universitätsmedizin Göttingen
Department of Neuro- and Sensory Physiology
Humboldtallee 23
Göttingen 37073
Germany

Dear Dr. Bonnin,

Thank you for submitting your revised manuscript entitled "High-resolution analysis of bound Ca²⁺ in neurons and synapses". We would be happy to publish your paper in Life Science Alliance pending final revisions necessary to meet our formatting guidelines.

- please address Reviewer 3's remaining comments
- please add ORCID ID for secondary corresponding -- he should have received instructions on how to do so
- please add the Twitter handle of your host institute/organization as well as your own or/and one of the authors in our system
- please remove figures from the main manuscript text
- please add your main, supplementary figure, and table legends to the main manuscript text after the references section
- please add callouts for Figures 1A; 2B; S1A-B; S2A-C; S3A-C; S5A-C; S7A-B; S8A-C to your main manuscript text
- please include a statement that approval was granted for the work with rat pups, and who provided this approval. This can be added in the "Hippocampal Cultures" section.
- the files uploaded as "Data S1-S3" appear to be Source Data. If so, please upload them as Source Data and label them so that it is clear which figure each file of source data corresponds to.
- Data S4 can be left as Supplemental Material, but label it "Matlab Scripts". please include the legend under the supplementary figure legends in the main manuscript
- the "Analytical solution of computational model" section from the Supplemental Material file should be incorporated into the main Materials and Methods section. Same for the References in this file.

A. FINAL FILES:

B. MANUSCRIPT ORGANIZATION AND FORMATTING:

Sincerely,

Reviewer #1 (Comments to the Authors (Required)):

This manuscript describes a new capability, the measurement of bound calcium, using a NanoSIMS. The revisions made to the manuscript make it more accessible to a broad audience and also clarify the significance of their findings. Therefore they have addressed my concerns, and I recommend the manuscript for publication.

Reviewer #2 (Comments to the Authors (Required)):

The authors describe a method to measure protein-bound calcium (Ca^{2+}). They propose nanoscale secondary ion mass spectrometry (NanoSIMS) as a new imaging tool to correlate bound Ca^{2+} to biological processes, such as synaptic activity. NanoSIMS analyses can be combined with fluorescent imaging of distinct compartments as well as dynamic assessment of free Ca^{2+} .

I appreciate the efforts of the authors to address most of my concerns. Based on their new experimental evidence, I support the publication of their manuscript. Congrats.

Reviewer #3 (Comments to the Authors (Required)):

The authors have clarified many of my concerns and I believe the conclusions of the paper are better supported now.

Just a minor comment - in figure 4B the scale of the lut goes from 0 to 255 - this seems confusing when comparing to the results of the correlations, where the Ca^{40} counts never reach anywhere over 70 counts. The image would make one think that 255 can be reached, but that does not seem to happen. It would be easier to understand if the lut scale was matched to actual counts, so one can get a sense of how the image and the spots found represent data points of the correlation.

Another small comment is that, while those correlations are statistically significant in figure 4, much of that seems driven by

outlier regions of high ^{40}Ca counts (which correspondingly have low antibody marked for syt and NB). It could generate concerns in the readership that the claim is too strong for data that is not equally sampled across the ^{40}Ca spectrum, thus not convincing fully on those regions of interest with high bound Ca. Would the authors consider claiming less strongly that Ca bound anti-correlates with activity? i.e. like in the abstract? This is not an effort on shaping the manuscript to my taste, but just helping to calibrate the claims as best as possible. That said, they authors should feel free to proceed as they want with this particular comment, it is simply a suggestion that it would appear that an important conclusion is drawn from relatively weak data.

Otherwise, I am happy to see in this manuscript the effort on pursuing such a complicated issue and I think this paper will be important in starting the discussion about the roles of bound Ca^{2+} for synapse physiology. I recommend its publication and I wont need to review again further.

Reply to the Reviewer Comments

All Reviewer comments are shown in *italics* below, with our answers in normal font.

Reviewer # 1

This manuscript describes a new capability, the measurement of bound calcium, using g a NansSIMS. The revisions made to the manuscript make it mre accessible to a broad audience and also clarify the significance of their findings. Therefore they have addressed my concerns, and I recommend the manuscript for publication.

We thank the Reviewer for the comments, and for recommending the manuscript for publication.

Reviewer # 2

The authors describe a method to measure protein-bound calcium (Ca²⁺). They propose nanoscale secondary ion mass spectrometry (NanoSIMS) as a new imaging tool to correlate bound Ca²⁺ to biological processes, such as synaptic activity. NanoSIMS analyses can be combined with fluorescent imaging of distinct compartments as well as dynamic assessment of free Ca²⁺.

I appreciate the efforts of the authors to address most of my concerns. Based on their new experimental evidence, I support the publication of their manuscript. Congrats.

We thank the Reviewer for the comments, and for recommending the manuscript for publication.

Reviewer # 3

The authors have clarified many of my concerns and I believe the conclusions of the paper are better supported now.

We thank the Reviewer for the comments.

Just a minor comment - in figure 4B the scale of the lut goes from 0 to 255 - this seems confusing when comparing to the results of the correlations, where the Ca40 counts never reach anywhere over 70 counts. The image would make one think that 255 can be reached, but that does not seem to happen. It would be easier to understand if the lut scale was matched to actual counts, so one can get a sense of how the image and the spots found represent data points of the correlation.

Because the LUT scale in this figure is meant to represent both the ⁴⁰Ca image and the ¹⁴N image, we cannot set the scale to the maximum counts for ⁴⁰Ca. However, we do recognize that using 255 may be confusing. We have adjusted the LUT scale to go from 0 to max, and applied this change to all figures where the LUT scale is present.

Another small comment is that, while those correlations are statistically significant in figure 4, much of that seems driven by outlier regions of high 40Ca counts (which correspondingly have low antibody marked for syt and NB). It could generate concerns in the readership that the claim is too strong for data that is not equally sampled across the 40Ca spectrum, thus not convincing fully on those regions of interest with high bound Ca. Would the authors consider claiming less strongly that Ca bound anti-correlates with activity? i.e. like in the abstract? This is not an effort on shaping the manuscript to my taste, but just helping to calibrate the claims as best as

possible. That said, they authors should feel free to proceed as they want with this particular comment, it is simply a suggestion that it would appear that an important conclusion is drawn from relatively weak data.

We appreciate the comment of the Reviewer, but this is only based on a visual impression of the graph, as the Reviewer acknowledges “much of that seems driven by outlier regions of high ^{40}Ca counts”. In fact, this visual impression is false, and the correlation is not driven by outlier regions of high Ca^{2+} counts.

A statistical analysis indicates that only the top 4.99% of the Ca^{2+} values can be seen as outliers (using SigmaPlot 10.0, from Systat, Inc.). Removing these values leaves the correlations highly significant ($p \leq 0.0001$).

To test whether our correlations are driven by high, but not necessarily outlier, values, we performed further statistical tests. We found the following:

The correlation between calcium and synaptotagmin antibodies remains significant even if we remove the top half of the calcium intensities:

- original correlation: $R = -0.154$, $P < 0.0001$ (significant)
- remove highest 10%: $R = -0.186$, $P < 0.0001$ (significant)
- remove highest 30%: $R = -0.2437$, $P < 0.0001$ (significant)
- remove highest 50%: $R = -0.217$, $P = 0.0009$ (significant)

The correlation between calcium and anti-mouse nanobodies behaves similarly, although this correlation is not as strong.

- original correlation: $R = -0.2194$, $P < 0.0001$ (significant)
- remove highest 10%: $R = -0.1223$, $P = 0.0013$ (significant)
- remove highest 30%: $R = -0.0980$, $P = 0.0236$ (significant)
- remove highest 50%: $R = -0.0426$, $P = 0.4074$ (not significant)

These calculations indicate that our conclusions are not based on any outlier effects. It is therefore not necessary to change the text or the interpretations.

Otherwise, I am happy to see in this manuscript the effort on pursuing such a complicated issue and I think this paper will be important in starting the discussion about the roles of bound Ca^{2+} for synapse physiology. I recommend its publication and I wont need to review again further.

We thank the Reviewer for recommending the manuscript for publication.

October 2, 2023

RE: Life Science Alliance Manuscript #LSA-2023-02030-TRR

Dr. Elisa A. Bonnin
Universitätsmedizin Göttingen
Department of Neuro- and Sensory Physiology
Humboldtallee 23
Göttingen 37073
Germany

Dear Dr. Bonnin,

Thank you for submitting your Methods entitled "High-resolution analysis of bound Ca²⁺ in neurons and synapses". It is a pleasure to let you know that your manuscript is now accepted for publication in Life Science Alliance. Congratulations on this interesting work.

DISTRIBUTION OF MATERIALS:

Again, congratulations on a very nice paper. I hope you found the review process to be constructive and are pleased with how the manuscript was handled editorially. We look forward to future exciting submissions from your lab.

Sincerely,
